# LuSh-NeRF: Lighting up and Sharpening NeRFs for Low-light Scenes

**Zefan Qu**     **Ke Xu**[†]     **Gerhard Petrus Hancke**     **Rynson W.H. Lau**[†]
Department of Computer Science
City University of Hong Kong
`zefanqu2-c@my.cityu.edu.hk, kkangwing@gmail.com,`
`{gp.hancke, Rynson.Lau}@cityu.edu.hk`

## Abstract

Neural Radiance Fields (NeRFs) have shown remarkable performances in producing novel-view images from high-quality scene images. However, hand-held low-light photography challenges NeRFs as the captured images may simultaneously suffer from low visibility, noise, and camera shakes. While existing NeRF methods may handle either light or motion, directly combining them or incorporating additional image-based enhancement methods does not work as these degradation factors are highly coupled. We observe that noise in low-light images is always sharp regardless of camera shakes, which implies an implicit order of these degradation factors within the image formation process. This inspires us to explore such an order to decouple and remove these degradation factors while training the NeRF. To this end, we propose in this paper a novel model, named LuSh-NeRF, which can reconstruct a clean and sharp NeRF from a group of hand-held low-light images. The key idea of LuSh-NeRF is to sequentially model noise and blur in the images via multi-view feature consistency and frequency information of NeRF, respectively. Specifically, LuSh-NeRF includes a novel Scene-Noise Decomposition (SND) module for decoupling the noise from the scene representation and a novel Camera Trajectory Prediction (CTP) module for the estimation of camera motions based on low-frequency scene information. To facilitate training and evaluations, we construct a new dataset containing both synthetic and real images. Experiments show that LuSh-NeRF outperforms existing approaches. Our code and dataset can be found here: https://github.com/quzefan/LuSh-NeRF.

## 1   Introduction

Neural Radiance Fields (NeRFs) [31, 2, 3, 5, 24] have achieved notable success in modeling 3D scene information via implicit functions learned from a set of 2D images with known camera poses. Since optimizing NeRFs by measuring the colorimetric errors of training views essentially requires bright and sharp training images, hand-held low-light photography, which is prevalent in our daily life (*e.g.*, in nighttime scenes), cannot be used directly for training NeRFs to produce visually pleasing novel view images (Fig. 1(b)), due to the co-existence of low visibility, noise, and camera motion blur in the captured images.

A straightforward solution is to incorporate existing low-light enhancement/deblurring methods (*e.g.*, [57, 12, 67, 59, 13, 45]) to preprocess the captured images before using them for training the NeRFs. However, it raises two problems. First, as these methods are typically image-based, they do not consider the multi-view consistency. Second, These enhancement methods may introduce

---

[†] Joint corresponding authors.

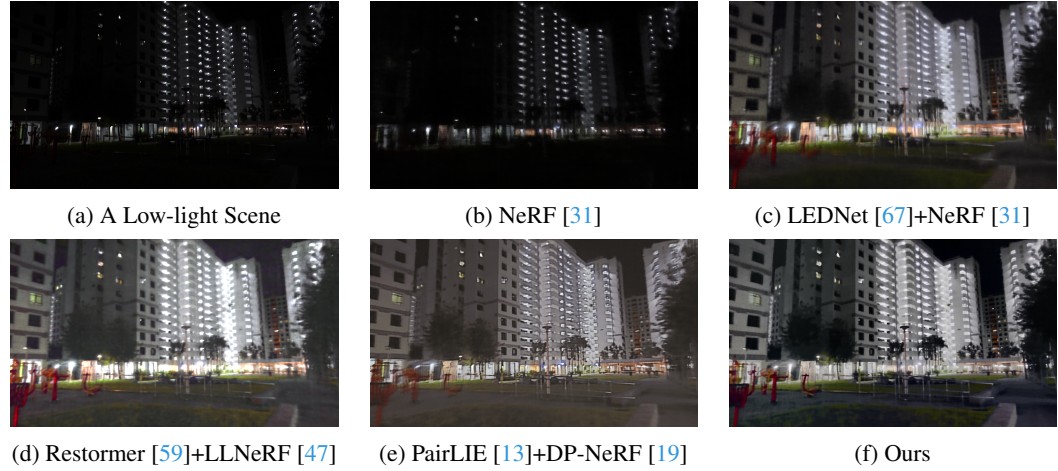

| (a) A Low-light Scene | (b) NeRF [31] | (c) LEDNet [67]+NeRF [31] |
| (d) Restormer [59]+LLNeRF [47] | (e) PairLIE [13]+DP-NeRF [19] | (f) Ours |

Figure 1: Given a hand-held captured low-light scene (a), while (a combination of) existing low-light enhancement/NeRF methods may not produce visually pleasing novel-view images ((b)-(e)), our LuSh-NeRF can produce bright and sharp results (f).

additional artifacts (*e.g.*, overexposure and unnatural color contrast). We note that there are some NeRF methods proposed for handling low-light scenes [47, 6] and scene motions [26, 48, 19, 7]. However, while the former assumes no camera motions occur during the capture, the latter cannot handle low-light scenes. Directly applying them to our problem does not work. A visual example is shown in Fig. 1((c) to (e)), where existing methods struggle to render the desired results.

We observe that in the captured low-light images, noise always appears sharp regardless of the camera shakes, due to the independent sensor noise generation within the collection and transformation of photons into electronic signals in the camera Image Signal Processor (ISP). This implies an implicit order of low visibility, sensor noise, and blur, which inspires us to model such an implicit order to decouple and remove those degradation factors for NeRF's training in an unsupervised manner.

In this paper, we propose a novel method, called LuSh-NeRF, to **L**ight **u**p and **Sh**arpen NeRF by sequentially modeling the degradation factors. Specifically, the brightness of training images is first scaled up to provide more contextual information (which simultaneously amplifies the noise in the images). We then propose two novel modules, *i.e.*, a Scene-Noise Decomposition (SND) module and a Camera Trajectory Prediction (CTP) module, to handle the noise and camera shake problems. The SND module learns to decouple noise from the implicit scene representation by explicitly learning a noise field through leveraging the multi-view feature consistency of NeRF. The CTP module then estimates the camera trajectories for sharpening image details based on the low-frequency information of denoised scene images rendered by SND. The two modules are optimized in an iterative manner such that the denoised scene representation of the SND module provides more information for predicting trajectories in the CTP module, while the results with sharp details from the CTP module in turn facilitate the denoising process in the SND module. To facilitate model training and evaluations, we construct a new dataset consisting of five synthesized scenes and five real scenes. As shown in Fig. 1(f), our LuSh-NeRF can render bright and sharp novel-view results.

In summary, this work has the following main contributions:

- We propose the first method (LuSh-NeRF) to reconstruct a NeRF from hand-held low-light photographs, by decoupling and removing degradation factors through modeling their implicit orders.

- Our LuSh-NeRF contains two novel modules, a novel SND module for noise removal from the implicit scene representation and a novel CTP module for handling camera motions.

- We construct the first dataset for training and evaluations. Experiments show that LuSh-NeRF outperforms existing methods.

## 2 Related Works

**Neural Radiance Field (NeRF).** NeRF [31] has fundamentally changed the way of modeling 3D scenes by learning coordinate-based implicit neural representations from a set of 2D observations. has gained widespread popularity in computer vision and graphics tasks pertaining to neural rendering, leveraging coordinate-based implicit neural representations (INR). It has gained significant popularity in computer vision and graphics tasks, with many NeRF variants proposed for, *e.g.*, accelerating the training and rendering of NeRF [14, 18, 40, 41], handling dynamic scenes [21, 34, 39, 44, 61] and digital humans body [1, 34, 36, 37, 10] or human head [53, 68, 16] modeling, and the manipulation [4, 38, 28, 23] or generation [8, 15, 25, 33] of scene contents. Many variants of NeRF are also gradually expanding into multiple research areas. For example, [14, 18, 40, 41] are proposed to accelerate the NeRF training procedure, [21, 34, 39, 44, 61] are applied to render dynamic scenarios, [4, 38, 28, 23] are focused on the NeRF relighting methods, [1, 34, 36, 37, 53] are expanded to the non-rigid object rendering, [8, 15, 25, 33] are used for the generation models. These methods typically require high-quality (*i.e.*, bright and sharp) 2D images for optimizing NeRFs.

**NeRF from Low-light Images.** Recently, there are some methods [47, 30, 6] proposed to relax such constraints by learning to reconstruct NeRFs from low-light images. Wang *et al.* [47] reconstructs a normally illuminated scene with multiple low-light images in an unsupervised manner. Cui *et al.* [6] can change the luminance of the scene by extending the transmittance function in NeRF. However, these methods do not consider the camera shakes that often occur in hand-held low-light imaging.

**NeRF from Blurry Images.** Rendering scenes with multiple blurry images is frequent and challenging in the real world. Ma *et al.* [26] firstly propose the deblurring problem in NeRF and simulate the blurring process with a deformable kernel. Most works [19, 48, 20, 17] model the camera motion trajectory in 3D space to handle the blur in normal-light scenes. Peng *et al.* [35] accelerate the deblurring process with an efficient rendering scheme. Huang *et al.* [17] utilize a blur generator to model the camera imaging process for the all-in-focus photos. However, these methods cannot handle hand-held low-light photographs as they do not jointly model noise and camera motions under low-light conditions.

**Low-light Image Enhancement.** Deep learning-based methods have been shown to be more effective in this task. Some Retinex-based methods [52, 65, 64, 54] and end-to-end methods [9, 11, 27, 32, 50, 56, 66, 58, 46, 22] are capable of achieving good enhanced results. Dong *et al.*[9] abandon the bayer-filter to recover the low light color from raw camera data. Dudhane *et al.*[11] merge sequential images taken in the same scene to enhance the image quality. Xu *et al.*[56] take an SNR prior to guide the feature learning and formulate the network with a new self-attention model. However, there are few methods that can address the blur present in practical low-light images owing to the long exposure time. Recently, Zhou *et al.*[67] propose a large dataset for the low-light deblurring task, and some works [67, 62] propose solutions to enhance low-light blurred images.

**Low-light Deblurring.** There are few methods that can address the blur present in practical low-light images owing to the long exposure time. Recently, Zhou *et al.*[67] propose a large dataset for the low-light deblurring task, and some works [67, 62] propose solutions to enhance low-light blurred images. Zhou *et al.* [67] propose a highly coupled encoder-decoder architecture to solve the joint artifact. Zhang *et al.* [62] delves into the multiple degradations and proposes an all-in-one fashion restoration network. While our work addresses a similar problem, we aim for NeRF reconstruction, which is more challenging.

## 3 Proposed Method

In this section, we will introduce the details of the proposed method LuSh-NeRF, which can reconstruct a normally illuminated, sharp, and clean scenario from a set of hand-held captured low-light 2D images. Specifically, we first propose the SND module to separate noise from the scene information in the pre-processed image (Sec. 3.2), and then use the CTP module to make an accurate prediction of the kernel from the low-frequency information in the image (Sec. 3.3). Our network is supervised by multiple loss functions (Sec. 3.4). The overall structure of the network is shown in Fig. 2.

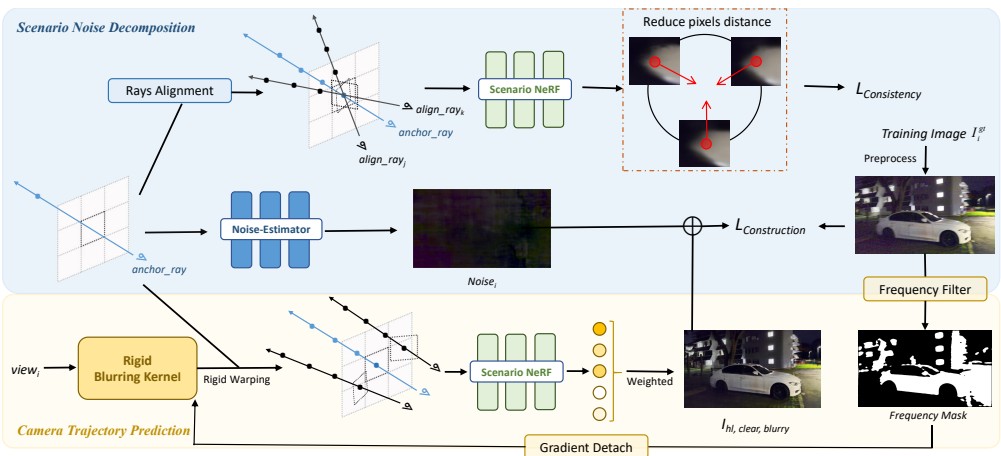

Figure 2: The pipeline of our proposed LuSh-NeRF. It contains two novel modules: (a) SND module: Decompose the noise in each view from the origin training image with a Noise NeRF architecture, and utilize the multi-view consistency characteristic in 3D scenario to separate the scene information and noise better; (b) CTP module: To minimize the interference of noise in low-light images on blur kernel predictions, the high frequency domain of the low light regions which are severely affected by noise are abandoned. In the rendering stage, we discard the Noise Estimator and Blur Kernel, and only use the Scenario-NeRF to render the enhanced scene.

## 3.1 Problem Statement

**Preliminary of Neural Radiance Field.** NeRF utilizes volume rendering [29] to synthesize 3D scenes by simulating multiple camera rays $\mathbf{r}(t) = \mathbf{o} + t\mathbf{d}$ emitted into the scene, where $\mathbf{o}$ represents the ray origin, $t$ is the sample distance, and $\mathbf{d}$ denotes the ray direction. The values at various sample points along these rays are encoded within an implicit neural network. Given a 3D coordinate $\mathbf{x} = (x, y, z)$ and a querying view direction $\mathbf{d} = (\theta, \phi)$, NeRF estimates the function $F : (\mathbf{x}, \mathbf{d}) \to (\mathbf{c}, \sigma)$ through an MLP network, where $\mathbf{c}$ and $\sigma$ represent the RGB color value and the volume density, respectively. After obtaining the information of each sample point on a ray, the pixel value $\hat{C}(\mathbf{r})$ of ray $\mathbf{r}$ can be computed as:

$$\hat{C}(\mathbf{r}) = \sum_{i=1}^{N} w_i \mathbf{c}_i = \sum_{i=1}^{N} T_i \cdot (1 - \exp(-\sigma_i \delta_i)) \cdot \mathbf{c}_i, \quad T_i = \exp\left(-\sum_{j=1}^{i-1} \sigma_i \delta_i\right), \quad (1)$$

where $\delta$ is the distance between two subsequent sample points on a ray. $T_i$ is the accumulated transmittance value that denotes the radiance decay rate of sampled points.

**Problem Modeling.** For images taken in practical low-light scenarios, degradation can be modeled into 3 categories, as shown in the example in Fig. 3: **(1) Low value intensity.** The pixel value intensities in the image are generally low, which leads to increased difficulty in NeRF training [47], resulting in poor results as shown in Fig. 1(b); **(2) Noise.** Due to the sensor noise generated within the collection and transformation of photons, the image pixel values are distorted randomly and unpredictably; and **(3) Camera motion blur.** Long exposure times inevitably lead to blur in the photo due to the camera motion. When trying to address one of the three defects, the others interfere significantly. Therefore, how to reconstruct a properly illuminated, clear, and sharp NeRF scene from this poor-quality image is a very challenging problem.

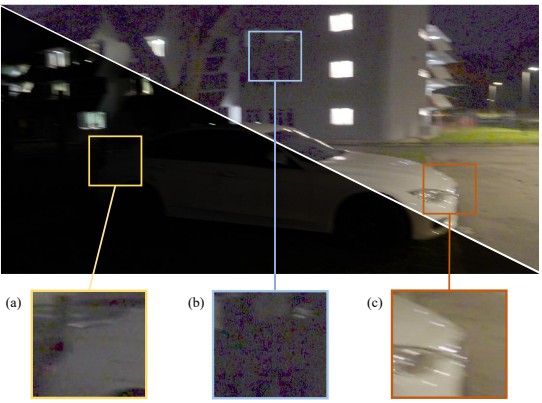

Figure 3: Different degradations in the real low light images. (a) Low intensity (b) Noise (c) Blur.

In this work, we attempt to optimize the NeRF by decoupling and removing the above defects in the training stage. Modeling and analyzing noise and blur are difficult at low pixel intensity values. Thus, the low visibility of the image should first be raised. Considering that noise in a low-light photo is mainly generated by the imaging process and would not be affected by the camera motions, the implicit order of the defect decomposition should be low visibility, noise, and blur.

Specifically, given multiple training images $I_{ll,noisy,blurry}$ from different views, NeRF aims to render a scenario with $I_{hl,clear,sharp}$. We first scale up the brightness of $I_{ll,noisy,blurry}$ and get $I_{hl,noisy,blurry}$ for more contextual information, but this process simultaneously amplifies the noise in the image. LuSh-NeRF then decomposes the image into scene and noise information with 3D multi-view consistency, removing the noise from the image to obtain $I_{hl,clear,blurry}$. Finally, the image without noise interference is utilized to model the blur phenomenon and restore $I_{hl,clear,sharp}$. The enhancement process for LuSh-NeRF is as follows:

$$I_{hl,clear,sharp} = \text{R(LuSh-NeRF)} = \text{Sharpen}(\underbrace{\text{ScaleUp}(I_{ll,noisy,blurry}) - \text{Noise}}_{I_{hl,clear,blurry}}), \qquad (2)$$

where Noise is the noise of the different views predicted by the SND module. $\text{Sharpen}(\cdot)$ denotes the CTP module to ease the camera motion blur phenomenon, and $\text{R}(\cdot)$ is the NeRF rendering function.

During training, due to the lack of ground truth, $I_{hl,noisy,blurry}$ (*i.e.*, $\text{ScaleUp}(I_{ll,noisy,blurry})$) is used as the supervised signal. The inverse of Eq. 2 should be taken to simulate the low-quality image as:

$$\hat{I}_{hl,noisy,blurry} = \text{Sharpen}^{-1}(\text{R}(LuSh-NeRF)) + \text{Noise}, \qquad (3)$$

After training, the correct-light (denoted as *hl*), sharp and clear NeRF model can be obtained directly by discarding the $\text{Sharpen}^{-1}$ function and Noise. In the following subsections, we introduce how we estimate the Noise under each view with the SND module and how we accurately predict the $\text{Sharpen}^{-1}$ function with the CTP module.

## 3.2 Scenario-Noise Decomposition

Directly training NeRF with $I_{hl,noisy,blurry}$ would still produce defective rendering results, as scaling up the low-light images can amplify noise and distort colors significantly, causing more severe multi-view inconsistency.

We note that while the noise is generally sharp and stochastic and has different values in different views, scenario information across views tends to be consistent. This makes it possible to decouple noise and scenario information. To co-optimize with the base network Scenario-NeRF (S-NeRF), we propose a new network, named Noise-Estimator (N-Estimator), which has the same network structure as S-NeRF, to compute the noise of each rendering coordinate. Since the noise is not consistent, N-Estimator discards the volume rendering calculation and directly takes the values of the intermediate sampling points of the input rays as the output. The formula for obtaining the noisy pixel $C_{noisy}(r)$ is:

$$C_{noisy}(r) = \text{CTP}(C_{S-NeRF}(r)) + C_{N-Estimator}(r) = \text{CTP}(\sum_{i=1}^{N} w_i \mathbf{c}_i) + n_{\frac{N}{2}}, \qquad (4)$$

$$where \quad n_{\frac{N}{2}} = MLP_{N-Estimator}(P_{mid}, d),$$

where $P_{mid}$ is the intermediate sampling points coordinate, and $d$ is the view direction. The $\text{CTP}(\cdot)$ function refers to the Camera Trajectory Prediction module, $N$ is the number of sampled points on a ray, and $n_{\frac{N}{2}}$ is the noise value rendered by N-Estimator. The volume rendering computation of S-NeRF gradually forces S-NeRF to learn the consistency scenario information, and the noise that does not share this feature is decomposed from the scene representation by the N-Estimator during the optimization process to fit the final noisy training images.

To more accurately decouple the scene information and noise, LuSh-NeRF utilizes the Rays Alignment supervision in the SND module. Specifically, An image matching method [43] is applied to the images rendered by S-NeRF after sharpening in each viewpoint to obtain a dense pixel matching matrix $M$ and a certainty matrix $C$, which represent the coordinate pairs of matching pixels in different views and the confidence level of each matching. The current input ray is then taken as

$anchor\_ray$, and the coordinates of its rendered pixels are used to find the matching pixels under other viewpoints in $M$. The confidence of the matching pairs in $C$ should be higher than the preset threshold $\theta$ to ensure that accurate matches can be obtained.

With these matching pixel coordinates, we can get the corresponding aligned rays $align\_ray_i$ under different viewpoints $view_i$. Ideally, the RGB colors computed by $anchor\_ray$ and $align\_ray_i$ should be close to identical owing to the scenario consistency. So a consistency loss $L_{consistency}$ is proposed to reduce the distance within these rendering pixels in S-NeRF to force the decoupling of scene information and noise, as:

$$L_{consistency} = \frac{1}{K}\sum_{i=1}^{K}\left|C_{S-NeRF}(r_i) - \frac{1}{K}\sum_{j=1}^{K}C_{S-NeRF}(r_j)\right|, \tag{5}$$

where $K$ is the number of training views. Constrained by $L_{consistency}$, the information extracted by S-NeRF has a higher viewpoint consistency, leading to a better decomposition of the scene information and noise.

### 3.3 Camera Trajectory Prediction

Motivated by [19], the CTP module employs the same thoughts to predict the rigid camera motions with all rays in each view. However, unlike the properly lighted blurry data used in [19], the practical low light conditions present substantial interferences, which could cause notable errors in the CTP network predictions, as demonstrated in the results of Subsec 4.1.

When the SND module is not fully converged, noise still exists in the SND-processed scenario. The kernel prediction using rays with strong noise interference may interrupt the CTP network, and directly lead to the failure of the whole NeRF training. Hence, it is necessary to filter the rays for CTP module optimization. Since the high-frequency and low-intensity regions in the low-light images are more disturbed by noise, corrupting the information in the original raw image [55], the CTP module utilizes Discrete Fourier Transform (DFT) to obtain the image frequency map as:

$$Y(u,v) = \text{DFT}(x(m,n)) = \sum_{m=0}^{M-1}\sum_{n=0}^{N-1}x(m,n)\cdot e^{-j2\pi\left(\frac{um}{M}+\frac{vn}{N}\right)}. \tag{6}$$

The noisy high-frequency regions of the image are filtered by the low-pass filter $H$ of radius $r$:

$$Y_{\text{lowpass}}(u,v) = H(u,v)\cdot Y(u,v), \quad \text{where } H(u,v) = \begin{cases} 1, & \text{if } \sqrt{u^2+v^2} \le r \\ 0, & \text{if } \sqrt{u^2+v^2} > r \end{cases}. \tag{7}$$

The retained low-frequency image regions are then converted back to the image via the $\text{DFT}^{-1}$ function to obtain the image informative region mask $Mask_{\text{lowpass}}(m,n)$ as:

$$Mask_{\text{lowpass}}(m,n) = \text{Bi}(\text{DFT}^{-1}\left[Y_{\text{lowpass}}(u,v)\right],T), \tag{8}$$

where $\text{Bi}(\cdot,T)$ is a binarization function with threshold $T$. $Mask_{\text{lowpass}}(m,n)$ classifies the rays into $ray_{clear}$ and $ray_{noisy}$, which are used to render the clear and noise-dominated pixels in the training images separately. The gradient of $ray_{noisy}$ to the rigid motion prediction network is detached during NeRF optimization to reduce the impact of noise yet to be removed by the SND module on blur kernel prediction as:

$$rays = [ray_{clear}, \text{Detach}(ray_{noisy})], \tag{9}$$

where the $\text{Detach}(\cdot)$ function is used to detach the gradient from the variables. The role of the low-pass frequency filter is shown in Fig. 7, where the CTP Mask reduces the interference of noisy regions on the blur kernel prediction compared to the mask obtained by directly using the RGB intensity as the threshold.

### 3.4 Training & Rendering

After employing the SND and CTP modules as in Eq. 3, we have the simulated RGB values $\hat{C}_{hl,noisy,blurry}$ in the image $\hat{I}_{hl,noisy,blurry}$. We can then optimize the network with the MSE loss $L_{consturction}$ as:

$$L_{construction} = \sum_{i}\|\hat{C}_{hl,noisy,blurry}(r_i) - C_{hl,noisy,blurry}(r_i)\|_2. \tag{10}$$

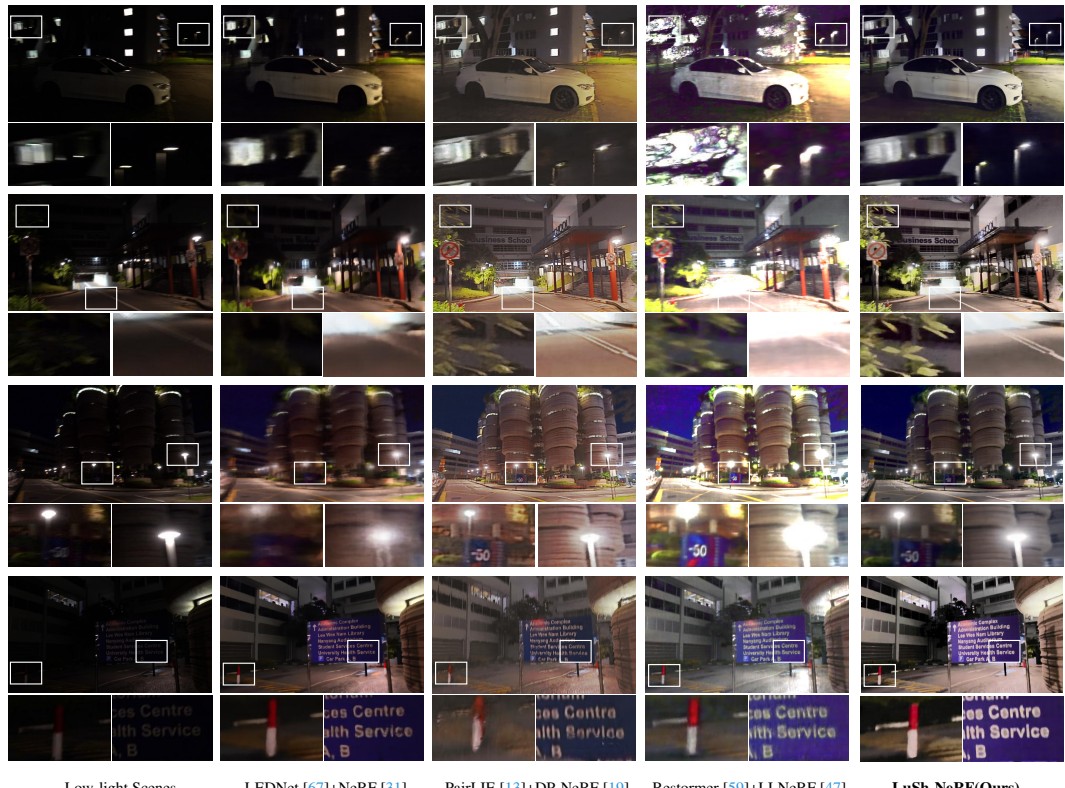

| Low-light Scenes | LEDNet [67]+NeRF [31] | PairLIE [13]+DP-NeRF [19] | Restormer [59]+LLNeRF [47] | **LuSh-NeRF(Ours)** |

Figure 4: Qualitative results of different methods on our real scenes. Our LuSh-NeRF can render cleaner and sharper results for real low-light scenes with camera motions.

All the modules in the network are trained in an end-to-end manner with the following training losses:

$$L_{LuSh-NeRF} = \alpha \cdot L_{construction} + \beta \cdot L_{consistency}, \tag{11}$$

where $\alpha$ and $\beta$ are the hyper-parameters to balance the impact of two losses. When rendering a scenario, only the S-NeRF network in LuSh-NeRF is utilized with the volume rendering method to produce images of different views.

## 4 Experiments

**Our Dataset.** Since we are the first to reconstruct NeRF with hand-held low-light photographs, we build a new dataset based on the low-light image deblur dataset [67] for training and evaluation. Specifically, our dataset consists of 5 synthetic and 5 real scenes, for the quantitative and generalization evaluations. We use the COLMAP [42] method to estimate the camera pose of each image in the scenarios. Each scenario contains 20-25 images, at $1120 \times 640$ resolution. The brightness of images in each scenario is extremely low, where the intensities of most pixels are below 50. 80% of the images contain camera shake problems, which pose a significant challenge for NeRF reconstruction. Refer to Appendix A.3 for more details of our dataset.

**Implementation Details.** We have implemented our LuSh-NeRF based on the official code of Deblur-NeRF [26], and with the same Rigid Blurring Kernel network in [19]. The S-NeRF shares the same structure as NeRF [31], while the the depth and width of N-Estimator are set to half of those of the S-NeRF. The number of camera motions $k$ and the frequency filter radius in the CTP module are set to 4 and 30. The number of aligned rays $K$ and certainty threshold $\theta$ in the SND module are set to 20 and 0.8. Before training, the input images are up-scaled by gamma adjustment and histogram equalization. The batch size is set to 1,024 rays, with 64 fine and coarse sampled points. $\alpha$ and $\beta$ are set to 1 and 0 during the first 60K iterations for better rendering results, to avoid the inaccuracy matching matrix $M$ interfering with the SND module. The two hyper-parameters are then changed to

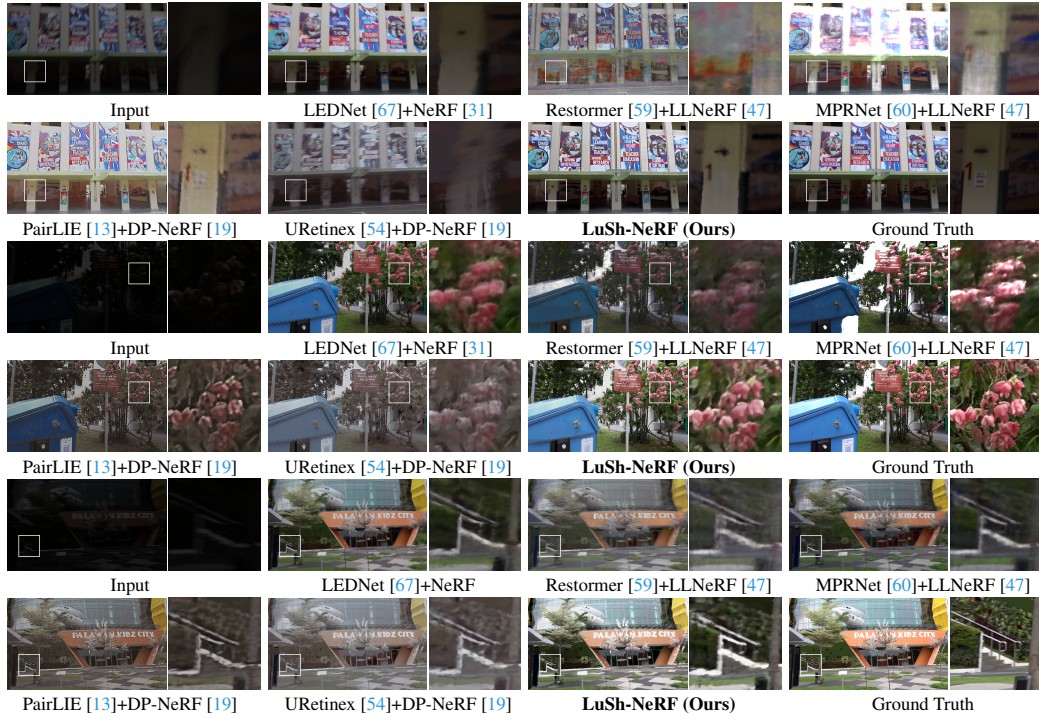

Figure 5: Qualitative results of different methods on our synthetic scenes. Our method yields the most natural restoration results while sharpening the image.

1 and $1 \times 10^{-2}$ in the last 40K iterations. All the experiments in this paper are performed on a PC with an i9-13900K CPU and a single NVIDIA RTX3090 GPU.

## 4.1 Main Results

**Evaluation Methods and Metrics.** Since we are the first to tackle the NeRF reconstruction from low-light images captured with camera motions, we design several baseline methods by combining existing state-of-the-art image-based and NeRF-based methods: (1) Low-light image enhancement [54, 49] + Image deblurring method [60, 59] + NeRF [31]; (2) Joint low-light image enhancement and deblurring method [67] + NeRF [31]; (3) Low-light image enhancement [13, 54] + deblurring NeRF [19]; and (4) Image deblurring [59, 60] + low-light NeRF enhancement [47]. All the selected models have been shown to achieve SOTA performances on their individual tasks. PSNR, SSIM, and LPIPS [63] metrics are used to evaluate the performance difference of the rendered images between LuSh-NeRF and the combination of other state-of-the-art methods. Due to the significant variations in color contrast ratio and image brightness of the restored low light images by different methods, we consider LPIPS (Learned Perceptual Image Patch Similarity), which can reasonably assess perceptual image quality and visual differences rather than the pixel-level difference, as the key metric.

**Visual Comparisons.** Fig. 4 presents qualitative comparison of the rendering results from the above three combined methods with those of our LuSh-NeRF in our real low-light scenarios. We can see that the results from LEDNet [67] are still dark, which significantly limits the learning capability of the NeRF. While PairLIE [13] combined with DP-NeRF [19] can reconstruct clear rendering results, they are rather unnatural. The color contrast is severely shifted, resulting in unrealistic images with some lost details and incomplete blur removal. In addition, we can also observe that first deblurring on low-light images and then applying the LLE method, as in Restormer [59] + LLNeRF [47], is less effective, and there is still substantial blurring in the results. In contrast, our approach can recover detailed scenes with natural colors, resulting in a sharp and clean novel view visualization.

Fig. 5 presents qualitative comparison between existing methods and our LuSh-NeRF on our synthetic scenes. All of the above methods suffer serious performance degradation when the image luminance is low and the camera shake is severe (*e.g.*, the bottom of the first example in Fig. 5), where LuSh-NeRF can still reconstruct a satisfactory scene by decoupling the defects of the scene. As we can see from

Table 1 (top half):

| Methods | "Dorm" PSNR↑ | SSIM↑ | LPIPS↓ | "Poster" PSNR↑ | SSIM↑ | LPIPS↓ | "Plane" PSNR↑ | SSIM↑ | LPIPS↓ |
|---|---|---|---|---|---|---|---|---|---|
| NeRF [31] | 6.02 | 0.0307 | 0.8030 | 11.25 | 0.5159 | 0.4061 | 5.53 | 0.0716 | 0.8418 |
| LLFormer [49] + MPRNet [60] + NeRF [31] | 19.01 | **0.5558** | 0.4571 | 14.67 | 0.5765 | 0.2785 | 18.99 | 0.5258 | 0.5290 |
| URetinexNet [54] + Restormer [59] + NeRF [31] | 17.47 | 0.5240 | 0.4820 | 12.38 | 0.3483 | 0.6347 | 17.06 | 0.5205 | 0.5256 |
| LEDNet [67] + NeRF [31] | 18.14 | 0.6025 | 0.3530 | 21.09 | 0.7140 | 0.3009 | 15.21 | 0.5869 | 0.4137 |
| Restormer [59] + LLNeRF [47] | 17.85 | 0.5541 | 0.5000 | 14.03 | 0.5462 | 0.4182 | 14.94 | 0.5250 | 0.5260 |
| MPRNet [60] + LLNeRF [47] | 17.27 | 0.5493 | 0.4942 | 10.88 | 0.4744 | 0.4334 | 13.47 | **0.5420** | 0.5146 |
| PairLIE [13] + DP-NeRF [19] | 14.08 | 0.4574 | 0.3651 | 13.34 | 0.4482 | 0.2995 | 13.11 | 0.5048 | 0.4316 |
| URetinexNet [54] + DP-NeRF [19] | 16.55 | 0.4884 | 0.3762 | 14.12 | 0.4199 | 0.5829 | 16.43 | 0.5211 | 0.4472 |
| LuSh-NeRF (Ours) | **19.06** | 0.5354 | **0.3491** | 18.12 | **0.6331** | **0.2265** | **19.34** | 0.5275 | **0.3852** |

| Method | "Sakura" PSNR↑ | SSIM↑ | LPIPS↓ | "Hall" PSNR↑ | SSIM↑ | LPIPS↓ | Average PSNR↑ | SSIM↑ | LPIPS↓ |
|---|---|---|---|---|---|---|---|---|---|
| NeRF [31] | 7.54 | 0.0553 | 0.7186 | 8.19 | 0.1679 | 0.5048 | 7.71 | 0.1683 | 0.6548 |
| LLFormer [49] + MPRNet [60] + NeRF [31] | 16.04 | 0.5204 | 0.4192 | **22.02** | **0.7153** | 0.3026 | 18.14 | 0.5787 | 0.3973 |
| URetinexNet [54] + Restormer [59] + NeRF [31] | 16.29 | 0.5152 | 0.4347 | 20.39 | 0.7045 | 0.2834 | 16.72 | 0.5225 | 0.4721 |
| LEDNet [67] + NeRF [31] | 18.24 | 0.6127 | 0.2772 | 19.33 | 0.7279 | 0.2570 | 18.40 | 0.6488 | 0.3204 |
| Restormer [59] + LLNeRF [47] | 15.22 | 0.5016 | 0.4199 | 20.04 | 0.7126 | 0.3452 | 16.42 | 0.5679 | 0.4419 |
| MPRNet [60] + LLNeRF [47] | 14.25 | 0.5155 | 0.3923 | 14.16 | 0.5213 | 0.4958 | 14.01 | 0.5205 | 0.4661 |
| PairLIE [13] + DP-NeRF [19] | 13.86 | 0.5137 | 0.3157 | 19.65 | 0.6055 | 0.2600 | 14.81 | 0.5059 | 0.3344 |
| URetinexNet [54] + DP-NeRF [19] | 16.38 | 0.5286 | 0.3550 | 20.21 | 0.6328 | 0.2616 | 16.74 | 0.5182 | 0.4046 |
| LuSh-NeRF (Ours) | **18.94** | **0.5884** | **0.2562** | 21.09 | 0.6421 | **0.2400** | **19.31** | 0.5853 | **0.2914** |

Table 1: Quantitative comparison of different methods on our synthetic scenes. The **best** and the second performances of each scenario are marked in the table.

the given visualizations on realistic and synthesized data, higher PSNR and SSIM metrics of the methods (*e.g.*, LEDNet + NeRF) do not entirely represent better performances on the NeRF results in our proposed task. This demonstrates that the perceptual metric, LPIPS, is better at assessing the merits of the reconstructed scenario. Refer to Appendix A.5 for more visual results.

**Quantitative Comparison.** We conduct a quantitative comparison of our method against various combinations of SOTA approaches on our synthesized data in Tab. 1. Note that our synthesized data originates from the training and testing sets of LOL-Blur dataset [67], while the weight of LEDNet [67] is also specifically trained on the synthesized dataset of LOL-Blur. This condition does not constitute a fair comparison. Nonetheless, we report the results of using "LEDNet [67] + NeRF [31]" as a reference.

Our method achieves better performances than multiple combinations of existing methods on all 5 synthesized data. In terms of the LPIPS metric on all the scenarios, our method outperforms the rendering results of LEDNet [67] + NeRF [31] in an unsupervised optimization manner, which proves that the SND and CTP modules in LuSh-NeRF fully utilize the 3D information of the scene to aid in the recovery of the irreversible quality degradation in the single image.

**Ablation Study.** Fig. 6 demonstrates the effect of the various components of LuSh-NeRF on a realistic scenario. After the images are preprocessed to enhance the luminance, the noise in the images is significantly amplified (as shown in the white box). Directly using blur kernel prediction [19] on these images is heavily interfered by noise, which prevents recovery from camera shakes (Fig. 6(b)). In addition, incorrect kernel predictions have a negative impact as they make it difficult to align the multi-view images rendered by S-NeRF and the SND module cannot obtain accurate viewpoint consistency supervision (Fig. 6(c)). When the CTP module is utilized, the blur of the image is alleviated, but the noise problem is still serious (Fig. 6(d)). First sharpening the image with CTP

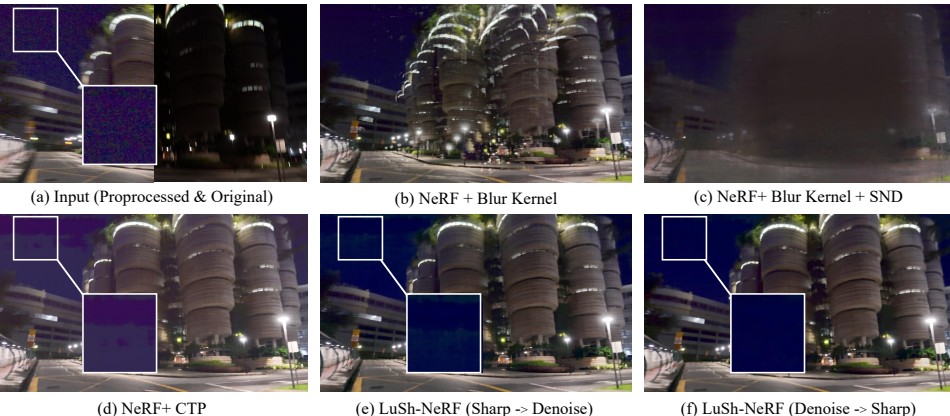

(a) Input (Proprocessed & Original)  (b) NeRF + Blur Kernel  (c) NeRF+ Blur Kernel + SND
(d) NeRF+ CTP  (e) LuSh-NeRF (Sharp -> Denoise)  (f) LuSh-NeRF (Denoise -> Sharp)

Figure 6: Ablation study of LuSh-NeRF on a real scenario.

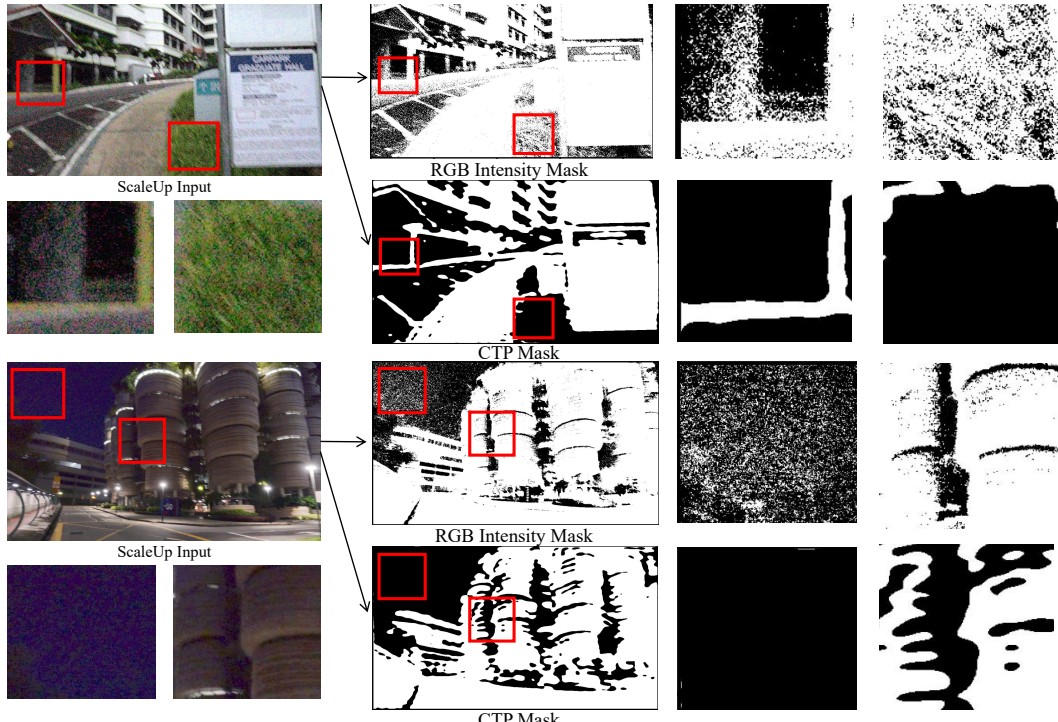

Figure 7: The RGB intensity mask and CTP mask comparison. Two masks are generated with the same threshold $T$. White points represent 1 and black points represent 0 in each mask.

and then removing noise with SND can somewhat suppress the noise problem, but the noise may be reconstructed to the various viewpoints and is not completely removed (Fig. 6(e)). LuSh-NeRF can render sharp and clean scenario images in novel views after the training process (Fig. 6(f)).

We show in the Fig. 7 the difference between the two masks obtained by directly performing an RGB intensity threshold and performing a frequency filtering before taking a threshold. (1) The RGB intensities in many noise-dominant regions are also high (e.g., the sky and the grass), which are harmful to the blur estimation but not excluded in the mask produced by thresholding. (2) The CTP module uses a frequency filter to identify the low-frequency-dominated regions of the image, and then obtains the desired mask based on the RGB intensity values. The gradients of rays in high-frequency and dark regions (regions more severely affected by noise) are detached during the Blur Kernel optimization process, which ensures better blur modeling.

## 5 Conclusion and Limitation

In this paper, we have proposed the NeRF reconstruction task from low-light images with camera motions, and a novel unsupervised reconstruction network LuSh-NeRF. LuSh-NeRF models the NeRF restoration process based on the characteristics of three kinds of defects in the scene, *i.e.*, low-light, noise, and camera blur, and successfully reconstructs a normal-light, clean, and sharp NeRF network from poor-quality low-light images, by exploiting multi-view consistency and frequency-domain information. To facilitate the training and evaluation of the new proposed task, we construct a new dataset which contains 5 synthetic and 5 real scenes low light images with hand-held camera motions. Extensive experiments on our proposed dataset demonstrate that our method is capable of achieving satisfactory NeRF reconstruction results, despite the defects in the input images.

The LuSh-NeRF network also has some limitations. One limitation is that it needs to optimize two NeRF networks at the same time and render multiple rays for the blur problem when sharpening images during training, which results in a slower optimization process, even though this would not interfere with the rendering speed. Besides, noise that is relatively similar across views may be difficult to remove due to the reliance on viewpoint consistency. (Refer to Appendix A.8 for details.) As a future work, we would like to address these issues.

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

# A Appendix

## A.1 Broader Impacts

While we do not foresee our method causing any direct negative societal impact, it may be leveraged to help create fake images using image generation techniques. The human information in the captured images may have the risk of a leak that raises privacy concerns. We urge the readers to limit the usage of this work to legal use cases.

## A.2 Details of the RBK module

The Rigid Blurring Kernel (RBK) module proposed in the DP-NeRF [19] is utilized in our CTP module. It mimics the scene blur kernel by simulating the 3D camera motions, which contains the following two main parts:

**Ray Rigid Transformation**: The blurring process of an image is modeled by the Ray Rigid Transformation (RRT). The RRT is formulated as ray transformation derived from the deformation of rigid camera motion, defined as the dense SE(3) field for scene $s$, approximated by the MLPs as follows:

$$S_s = (r_s; v_s) = (\mathcal{R}(\mathcal{E}(l_s)); \mathcal{L}(\mathcal{E}(l_s))), where \ s \in view_{img}, \tag{12}$$

where $l_s$ is the latent code for each view through the embedding layer in [1], $\mathcal{R}, \mathcal{L}, \mathcal{E}$ are three MLP networks, $view_{img}$ is the training image set. The $S_s = (r_s; v_s) \in \mathbb{R}^6$ is the matrix which will be used for the RRT modeling as follows:

$$ray_{s;q}^{RRT} = Rigid\_Transform(ray_s, (r_s; v_s)), \tag{13}$$

where $ray_s$ and $ray_{s;q}^{RRT}$ are the orgin ray and the transformed rays in scene $s$, $q \in \{1, ..., k\}$, k s a hyper-parameter that controls the number of camera motions contributing to the blur in each scene $s$. The blurry RGB value at $ray_s$ can be acquired by weighting sums of the NeRF volume rendering values $C_{s;0}$ and $C_{s;q}$ from $ray_s$ and $ray_{s;q}^{RRT}$.

**Coarse Composition Weights**: This part is responsible for computing the weights of each ray obtained by the RRT part, which is given by the following equation:

$$w_{s;0,1,...,k} = \sigma(\mathcal{W}(\mathcal{E}(l_s))), where \ \sum_{i=0}^{k} m_{s;i} = 1, \tag{14}$$

where $m_s$ is the final weight for each ray in RRT. Finally, blurry color $C_s$ for scene $s$ can be computed by the weighted sum operation as shown below:

$$C_s = m_{s;0}C_{s;0} + \sum_{q=1}^{k} m_{s;q}C_{s;q}. \tag{15}$$

## A.3 Details of the Proposed Dataset

In this work, we collect low-light blurry images from five synthetic and real videos in the LOL-Blur dataset [67] and calculate the camera pose for each image with COLMAP [42] toolbox. For synthetic data, we use ground truth images to estimate camera pose, while for realistic data, the images recovered from LEDNet pre-trained model are utilized to estimate pose since COLMAP is hard to model the scenario with the origin low light images. Images with large shifts in each scenario are removed. The number of images for each scene and the training and evaluation viewpoint allocation are shown in Tab. 2.

| Scenario | Synthesized Dataset | | | | | Realistic Dataset | | | | |
|---|---|---|---|---|---|---|---|---|---|---|
| | "Dorm" | "Poster" | "Plane" | "Sakura" | "Hall" | "Campus" | "Highway" | "Signboard" | "Car" | "Neighborhood" |
| Collected Views | 21 | 21 | 21 | 21 | 20 | 20 | 20 | 21 | 21 | 25 |
| Training Views | 18 | 18 | 18 | 18 | 17 | 17 | 17 | 18 | 18 | 21 |
| Evaluation Views | 3 | 3 | 3 | 3 | 3 | 3 | 3 | 3 | 3 | 4 |

Table 2: The dataset split details for our proposed LOL-BlurNeRF dataset.

All images have a resolution of 1120 x 640. The mean value of all pixels in the images is low (less than 50) and camera motion blur is present in 80% of the images in each scene. As illustrated in Fig. 1, Fig. 4, Fig. 5 and Fig. 9, poor image quality in each scene leads to a challenging NeRF reconstruction task.

## A.4 Quantitative Ablation Studies

We have conducted detailed ablation studies on all the synthetic scenes in the Tab. 3.

| Scene | "Dorm" | | | "Poster" | | | "Plane" | | | "Sakura" | | | "Hall" | | | Average | | |
|---|---|---|---|---|---|---|---|---|---|---|---|---|---|---|---|---|---|---|
| | PSNR↑ | SSIM↑ | LPIPS↓ | PSNR↑ | SSIM↑ | LPIPS↓ | PSNR↑ | SSIM↑ | LPIPS↓ | PSNR↑ | SSIM↑ | LPIPS↓ | PSNR↑ | SSIM↑ | LPIPS↓ | PSNR↑ | SSIM↑ | LPIPS↓ |
| NeRF | 6.02 | .0307 | .8030 | 11.25 | .5159 | .4061 | 5.53 | .0716 | .8418 | 7.54 | .0553 | .7186 | 8.19 | .1679 | .5048 | 7.71 | .1683 | .6549 |
| Preprocess+NeRF | 19.09 | .5453 | .4675 | 19.07 | .7048 | .3088 | 19.66 | .5230 | .4986 | 18.73 | .5699 | .3666 | 21.34 | .6683 | .3124 | 19.58 | .6023 | .3908 |
| Preprocess+Rigid Blur Kernel | 18.89 | .5259 | .4353 | 17.23 | .5900 | .2805 | 19.13 | .5193 | .4185 | 18.23 | .5482 | .2789 | 20.43 | .6353 | .2684 | 18.78 | .5637 | .3363 |
| Preprocess+CTP | 18.52 | .5205 | .3654 | 17.02 | .5915 | .2415 | 19.32 | .5144 | .4048 | 18.27 | .5514 | .2715 | 20.25 | .6411 | .2577 | 18.68 | .5638 | .3082 |
| Preprocess+SND | **20.18** | **.5646** | .4390 | **20.94** | **.7385** | .2811 | **20.13** | **.5665** | .4873 | **19.16** | **.5889** | .3568 | **21.67** | **.7326** | .2801 | **20.42** | **.6382** | .3689 |
| Preprocess+Rigid Blur Kernel+SND | 18.99 | .5299 | .3630 | 18.05 | .6179 | .2598 | 18.93 | .5191 | .3954 | 18.65 | .5530 | .2752 | 20.74 | .6381 | .2434 | 19.07 | .5716 | .3074 |
| LuSh-NeRF (Sharp ->Denoise) | 18.66 | .5008 | .3514 | 17.38 | .5860 | .2600 | 19.13 | .5213 | .4076 | 18.24 | .5420 | .2589 | 20.72 | .6386 | .2667 | 18.83 | .5577 | .3089 |
| LuSh-NeRF (Denoise ->Sharp) | 19.06 | .5354 | **.3491** | 18.12 | .6331 | **.2265** | 19.34 | .5275 | **.3852** | 18.94 | .5884 | **.2562** | 21.09 | .6421 | **.2400** | 19.31 | .5853 | **.2914** |

Table 3: Ablation Study of different modules in our LuSh-NeRF. The **best** performances of each scenario are marked in the table.

(1) Lines 1 and 2 show that the ScaleUp preprocessing enhances the NeRF's reconstruction capabilities, resulting in an improved image.

(2) Lines 3 and 4 show that CTP leverages frequency domain information to refine Blur Kernel predictions, boosting the perceptual quality of the reconstructed images. However, this process may decrease the PSNR and SSIM scores as it neglects noise interference.

(3) From Lines 2 and 5, the SND module can disentangle noise and scene information from the input noisy-blurry images, leading to substantial improvements on the PSNR and SSIM metrics. However, the SND module is not capable of resolving the blur problem, which leads to a minor improvement in the image's perceptual quality (more important for the rendered images).

(4) The comparisons between Lines 4, 5, and 8 show that combining SND and CTP modules enhances image perceptual quality and outperforms using only CTP in terms of PSNR and SSIM scores.

(5) Lines 7 and 8 show that decoupling the noise in the scene first, and then modeling the scene blur is a more robust restoration order, as it can effectively reduce the interference of noise in the deblurring process for obtaining better performances.

We have conducted the ablation experiments of the CTP module on two synthetic scenes in the Tab. 4.

| Scene | "Dorm" | | | "Poster" | | | Average | | |
|---|---|---|---|---|---|---|---|---|---|
| | PSNR↑ | SSIM↑ | LPIPS↓ | PSNR↑ | SSIM↑ | LPIPS↓ | PSNR↑ | SSIM↑ | LPIPS↓ |
| No Threshold | 18.99 | 0.5299 | 0.3630 | 18.05 | 0.6179 | 0.2598 | 18.52 | 0.5739 | 0.3114 |
| RGB Threshold(T = 32) | **19.18** | 0.5308 | 0.3580 | 17.86 | 0.5896 | 0.245 | 18.52 | 0.5602 | 0.3015 |
| RGB Threshold(T = 48) | 18.71 | 0.4874 | 0.4455 | 17.92 | 0.6057 | 0.2418 | 18.32 | 0.5466 | 0.3437 |
| RGB Threshold(T = 64) | 18.62 | 0.4804 | 0.4474 | 17.53 | 0.5947 | 0.2420 | 18.08 | 0.5376 | 0.3447 |
| CTP Threshold(r = 10, T = 48) | 18.95 | 0.5052 | 0.3660 | 18.14 | 0.6306 | 0.2275 | 18.55 | 0.5679 | 0.2968 |
| CTP Threshold(r = 30, T = 32) | 19.02 | 0.5310 | 0.3515 | **18.23** | 0.6318 | 0.2373 | **18.63** | 0.5814 | 0.2944 |
| CTP Threshold(r = 30, T = 48)(Ours) | 19.06 | **0.5354** | **0.3491** | 18.12 | **0.6331** | **0.2265** | 18.59 | **0.5843** | **0.2878** |
| CTP Threshold(r = 50, T = 48) | 19.10 | 0.5051 | 0.3634 | 17.78 | 0.6114 | 0.2384 | 18.44 | 0.5583 | 0.3009 |

Table 4: The ablation studies of the CTP module.

The experimental results show that the frequency filter provides a more desirable mask, which is superior to the directly obtained mask with RGB intensity thresholds.

## A.5 More Visual Results

We provide more visual comparisons between our proposed LuSh-NeRF network and the sota methods combinations, *i.e.*, LEDNet [67] + NeRF, Restormer [59] + LLNeRF [47], MPRNet [60] + LLNeRF [47], PairLIE [13] + DP-NeRF [19] and URetinexNet [54] + DP-NeRF [19] in Fig. 9.

## A.6 Comparison with COLMAP-Free NeRF method

COLMAP-free NeRF methods robust to low-light and blur phenomena would be much more helpful for the problem that we propose. However, existing colmap-free NeRF methods may not handle our task easily. The table below compares to a COLMAP-Free NeRF [51].

| Scene | "Dorm" | | | "Poster" | | | "Plane" | | | Average | | |
|---|---|---|---|---|---|---|---|---|---|---|---|---|
| | PSNR | SSIM | LPIPS | PSNR | SSIM | LPIPS | PSNR | SSIM | LPIPS | PSNR | SSIM | LPIPS |
| Preprocess+NeRF(COLMAP) | 19.75 | 0.5599 | 0.4705 | 19.07 | 0.7048 | 0.3088 | 19.66 | 0.523 | 0.4986 | 19.49 | 0.5959 | 0.4260 |
| Preprocess+NeRF−− [51] | 18.95 | 0.5423 | 0.4762 | 19.13 | 0.6935 | 0.3341 | 19.62 | 0.5243 | 0.5129 | 19.23 | 0.5867 | 0.4411 |

Table 5: Performance comparison with COLMAP-free NeRF method.

The results demonstrate that the existing COLMAP-free NeRF method cannot effectively handle low-light scenes with motion blur, due to the inaccurate poses optimized from the image directly.

## A.7 The Generalizability of the GIM

The image-matching method used in LuSh-NeRF is GIM [43], which is a state-of-the-art generalizable image-matching model pre-trained with abundant Internet videos. As mentioned in Section 3.2, we also utilize a pre-defined threshold to select those high-confidence matches in the results produced by the GIM, avoiding low-quality matches that may affect the network's training negatively.

We show some of the matches obtained by GIM on S-NeRF rendered results in Fig. 8.

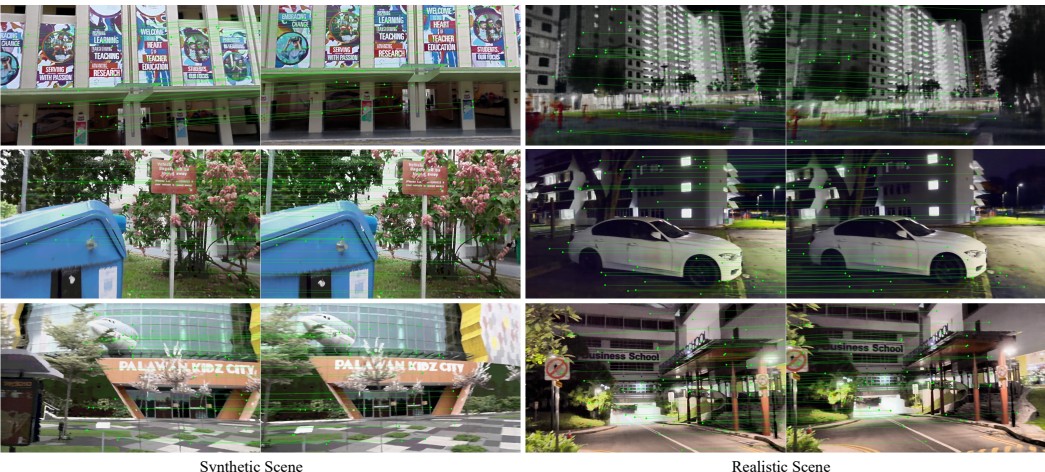

Synthetic Scene                                Realistic Scene

Figure 8: The visualization of matching results of GIM [43] on the S-NeRF rendering images.

## A.8 Discussion

LuSh-NeRF may have some limitations on certain data, as shown in Fig. 10. Although the noise in low light images is random and fluctuates greatly, if the noise in certain areas maintains similar characteristics in multiple viewpoints, LuSh-NeRF may model it as scene information, resulting in errors in rendering results. Increasing the training image numbers and spanning the range of different views may alleviate this problem to some extent.

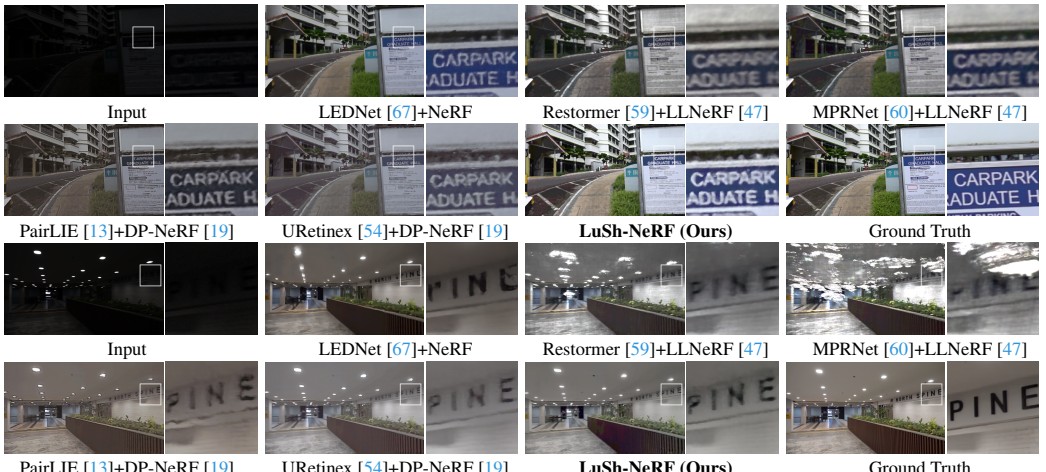

Figure 9: Qualitative results of different methods on the synthesized datasets.

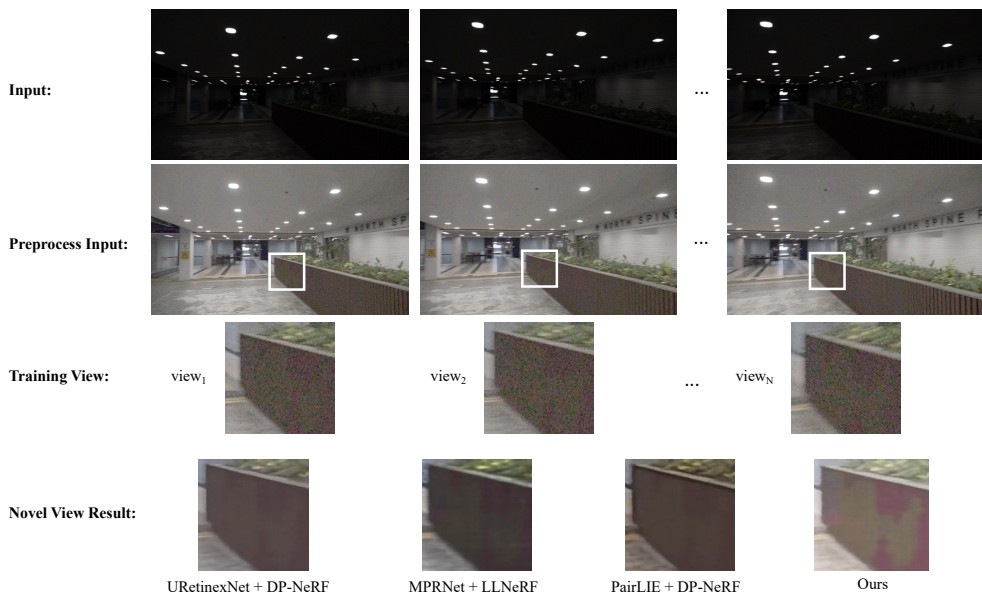

Figure 10: A limitation of the proposed LuSh-NeRF. Noise that is relatively similar across views may be difficult to remove due to the reliance on viewpoint consistency.

