# OpenReview forum: "LuSh-NeRF: Lighting up and Sharpening NeRFs for Low-light Scenes"
_NeurIPS.cc/2024/Conference — NeurIPS 2024 poster_

### Official Review · Reviewer_m3Lt · 2024-07-07

**Soundness:** 3
**Presentation:** 2
**Contribution:** 3
**Rating:** 6
**Confidence:** 4

**Summary:**

This work proposes a model to reconstruct a clean and sharp NeRF from a set of hand-held low-light images. The authors recognize the implicit order of the degradations (blur, noise, and low visibility), and sequentially decouple and remove each degradation in the network training. An SND module is proposed for noise removal and a CTP module is introduced for deblurring. They have conducted experiments on a dataset constructed from the LOL-Blur dataset proposed by [61].

**Strengths:**

1. This is the first work to reconstruct degradation-free images from a set of low-light photographs with motion blur.
2. They have introduced a new method for scene-noise decomposition from the implicit scene representation and utilized the frequency information for accurate kernel prediction NeRF.
3. The qualitative results are appealing and the authors have shown better quantitative scores for the method.

**Weaknesses:**

1. The presentation of the problem through mathematical equations is not clear. In Eq. 5, $C_{noisy}$ is the color value of a bright light blurry noisy (given in Eq. 4) image. But in Eq. 5, the first term $C_{S-NeRF}$ output is the sharp image and the second term is the noise term. The blurring operation is missing in this equation. Please correct me if I misinterpret anything. Also, N-NeRF does not use any volume rendering calculation to find the noise pixel value. Then, why this module is called a NeRF? It simply finds a pixel value for a given input pose. How the second term in Eq. 5 ($n\frac{N}{2}$) is derived? In Eq. 6, what are $i$ and $j$? In Fig. 2, the align ray is given subscripts $j$ and $k$. Please be consistent with equations and figure.

2. In the paper, it is given that the camera trajectory prediction (CTP) module is novel (Line 16, 47). But the same idea of [19] (Line 180) is used for camera trajectory prediction with the additional masking of high-frequency pixel regions in the image. The main function of deblur-NeRF in the CTP module is not novel.

3. Edge portions in the image will have high frequencies. When only rays containing low frequencies are used for gradient computation, typically edges (or high-frequency regions) will not come into the picture. But for deblur-NeRF, excluding the edge areas (high-frequency) is not a good idea since the blurring is mostly visible in the edges other than homogeneous (low-frequency) areas. How your frequency filter radius is selected? Did you consider any such factors for its selection?

4. In Line 224, it is given that for the first 60K iterations, $\beta$ is 0. This means SND does not come into the picture. In that case, the blur kernel estimated (by deblur-NeRF) will be wrong right (according to lines 185-186)? Is it correct to start from these trained weights after 60K iterations? What happens if $\beta$ is not 0 in the initial iterations?

5. SND relies on the image-matching method [40]. How generalizable is the method to your dataset? Have you taken the pre-trained weights?

6. The contribution of the dataset (given in Line 64) is not that significant since the poses of the available dataset from [61] are computed for the dataset generation using COLMAP. The method is tested on this single dataset (containing synthetic and real videos). Since the work is meant for handheld lowlight photography, a video of a low-light scene with handheld cameras (which typically will have a camera shake) could have been captured and the proposed method could have been tested on the dataset.

7. You have missed several references for Deblur NeRF papers. "ExBluRF: Efficient Radiance Fields for Extreme Motion Blurred Images", ICCV 2023; "PDRF: progressively deblurring radiance field for fast scene reconstruction from blurry images", AAAI 2023; "Inverting the Imaging Process by Learning an Implicit Camera Model", CVPR 2023. The related works (Sec. 2) section is not proper. There should be subsections for deblur NeRF and lowlight NeRF and briefly explain each work in both, other than explaining NeRF. Also, only low-light image enhancement methods are included. Why are deblurring methods not included? Or, it is better to have a subsection with explanations of lowlight deblurring works [61, 56]. Instead of just giving reference numbers, the essence of each very relevant work should be explained in a sentence in 'Related works'.

8. Some typos are there. In Line 153 and Fig. 6 (a) caption. In Line 35, [29] is not for low-light scenes. Line 41, the abbreviation of ISP is missing.

**Questions:**

Please see weaknesses.

**Limitations:**

Yes. Limitations are included in Section 5 and negative societal impacts are included in Section A.1.

---

> ### Author Rebuttal · Authors · 2024-08-07
>
> ### W1 Part1: Error in Eq.4 & 5 and the missing derivation.
> **R:** We thank the reviewer for the careful reading and apologize for the mistakes in Eq.5, which **omitted the deblurring process**.
>  The **missing derivation** is also added to Eq. 5. The revised Eq. 5 should be:
> $ C_{noisy}(r) = CTP(C_{S-NeRF}(r)) + C_{N-NeRF}(r) = CTP(\sum_{i=1}^{N} w_i c_i) + n_{\frac{N}{2}},
> \quad  where \quad n_{\frac{N}{2}} = MLP_{N-NeRF}(P_{mid}, d), $
>
> where the $P_{mid}$ and $d$ is the intermediate point coordinate and the view direction. The $\text{CTP}(\cdot)$ function is the CTP module, which is illustrated in the Subsec. 3.3.
>
> ### W1 Part2: Inproper name for N-NeRF.
> **R:** We refer to it N-NeRF as it is a **multilayer MLP network** similar to the S-NeRF, and it is **ray-conditioned** as the inputs are the view direction $d$ and the coordinates of the mid-point (fixed for each ray) during the whole training process. We will revise its name to **"Noise Estimator"** in the revision to avoid any ambiguities.
>
> ### W2: Insufficient novelty of CTP module.
> **R:** Note that DP-NeRF [19] does not consider the frequency domain information. DP-NeRF and other works directly uses **the rays of all image regions** to predict the camera trajectory, which is severely interfere by the low light noise (see Fig. 6(b) for details).
>
> The novelty of our CTP module lies in that it learns to identify **low-frequency-dominated regions** that are **more robust to noise** for kernel prediction. With these regions, LuSh-NeRF can significantly reduces the influence of noise on camera trajectory modeling. Some ablation experiments is in the table below.
> |Scene|"Dorm"|||"Poster"|||
> |-|-|-|-|-|-|-|
> ||PSNR|SSIM|LPIPS|PSNR|SSIM|LPIPS|
> |NeRF+DP-NeRF Blur Kernel|18.99|0.5299|0.3630|18.05|0.6179|0.2598|
> |NeRF+CTP|**19.06**|**0.5354**|**0.3491**|**18.12**|**0.6331**|**0.2265**|
>
> From the table, it can be derived that the CTP module that utilizes the help of **low frequency information** can be **better employed** in low light scene.
>
> ### W3: Excluding the edge areas.
> **R:** The reviewer is correct that blurring is more noticeable in high-frequency areas. However, in our task, noise may also significantly affect the images, making the high-frequency information **unreliable** for blur kernel predictions.
>
> Our CTP module is proposed to minimize the negative influence of these **low-quality** regions for blur modeling. As shown in **Fig.2 in the rebuttal PDF**, the **main edges** in the image can be effectively **preserved**, while the severely disturbed regions (e.g., the grass and the sky) are discarded in the optimization process.
>
> The tables below and in R3Q2 show the quantitative results of using different frequency filter radius $r$ and intensity threshold $T$, based on which we set the radius to 30. Will discuss more in the revision.
> |Scene|"Dorm"||| "Poster" |||
> |-|-|-|-|-|-|-|
> || PSNR|SSIM|LPIPS|PSNR|SSIM|LPIPS|
> |No Mask|18.99|0.5299|0.3630|18.05|0.6179|0.2598|
> |r=10,T=48|18.95|0.5052|0.3660|18.14|0.6306|0.2275|
> |r=30,T=32|19.02|0.5310|0.3515|**18.23**|0.6318|0.2373|
> |**r=30,T=48**|19.06|**0.5354**|**0.3491**|18.12|**0.6331**|**0.2265**|
> |r=50,T=48|19.10|0.5051|0.3634|17.78|0.6114|0.2384|
>
> ### W4: Correct to start training N-NeRF after 60K iters?
> **R:** When $\beta=0$, N-NeRF in SND module is still **co-optimized** with S-NeRF for noise estimation. When $\beta$ is turned on after 60K iterations, $L_{consistency}$ will **reinforce the consistency** across different views, which **further** facilitates the noise estimation of N-NeRF.
>
> If $L_{consistency}$ is used at the beginning of the training phase, the Image Matching method cannot accurately align the images due to the **low quality** of the images rendered by the S-NeRF in the early training stage, which may **deteriorate** the performance and extend the training time.
>
> We report the results of turning on $\beta$ at different iterations as references:
> |Scene|"Dorm"|||"Poster"|||
> |-|-|-|-|-|-|-|
> ||PSNR|SSIM|LPIPS|PSNR|SSIM|LPIPS|
> |No $L_{consistency}$|18.80|0.5007|0.3554|18.07|**0.6343**|0.2344|
> |10K Iter|18.94|0.5161|0.3584|18.02|0.6251|0.2353|
> |30K Iter|19.06|0.5195|0.3502|**18.15**|0.6291|0.2294|
> |60K Iter (Ours)|**19.06**|**0.5354**|**0.3491**|18.12|0.6331|**0.2265**|
>
> As shown in the table, adding $L_{consistency}$ at 60k training iteration yields better results.
>
> ### W5: The generalizability of the image-matching method.
> **R:** We empirically find the pre-trained GIM[40] has great generalization to cross-domain data. Some of the matches obtained by GIM on S-NeRF rendered results are shown **in Fig.3 in the rebuttal PDF** to demonstrate its generalizability.
>
> ### W6: Insufficient Dataset Contribution.
> **R:** Actually, we did **not just** take images from the LOL-Blur dataset and then ran an off-the-shelf SFM method to obtain the camera parameters. To build an effective dataset, we did the following works:
>
>  (1) **Scene selection**: we go through the whole LOL-Blur dataset to select scenes **featuring different environments** (covering indoor/outdoor situations) with **different camera motions and lighting.**
>
>  (2) **Image selection**: we manually select 20-25 (out of around 60) images per scene with relatively **high-quality** for the estimation of camera poses, and ensure their luminance is close to imitate the real shooting situation.
>
>  (3) **Camera Pose Estimation**: Note that the estimation of camera pose by COLMAP for low-light blurry images are often **unreliable**. We first repeat the estimation of COLMAP **30 times for each scene** to select the optimal pose result, and then **manually tune** it with our baseline NeRF model to improve the accuracy.
>
> Our experimental results show that the resulting dataset helps us learn the **robust LuSh-NeRF** for handling low-light blurry scenes. We will clarify these in the revision.
>
> ### W7&8: Missing references and typos.
> **R:** As suggested, we will cite and discuss these papers, and correct the typos in the revision.

---

> > ### Comment · Reviewer_m3Lt · 2024-08-09
> > **Clarification on W1Part1 response**
> >
> > Thank you for your response. Most of my concerns are addressed by the authors. But I have one small doubt in the response.
> >
> > In the updated Eq. 5 given in W1Part1 response, $MLP_{N-NeRF}(P_{mid}, d) = n N/2$. But Line 159 of the paper says that '$n$ is the noise value rendered by N-NeRF'. Which one is correct? $MLP_{N-NeRF}(P_{mid}, d) = n N/2$ or $MLP_{N-NeRF}(P_{mid}, d) = n$

---

> > > ### Author Response · Authors · 2024-08-09
> > > **Clarification on W1Part1**
> > >
> > > Thanks for the reply. We are glad that our rebuttal could address your concerns.
> > >
> > >  $n_{\frac{N}{2}} = MLP_{N-NeRF}(P_{mid}, d)$ is correct. We will correct $n$ into $n_{\frac{N}{2}}$ in Line 159 and check elsewhere to maintain the consistency of symbols in our revision.

---

> > > > ### Comment · Reviewer_m3Lt · 2024-08-09
> > > >
> > > > Thank you for the clarification. I am increasing the rating to 'weak accept'.

---

> > > > > ### Author Response · Authors · 2024-08-10
> > > > >
> > > > > Thanks very much for your scores and review comments, your detailed comments effectively help us to improve the quality of manuscript.
> > > > >
> > > > > We promise to incorporate all these missing details in the revision.

---

### Official Review · Reviewer_4eod · 2024-07-10

**Soundness:** 3
**Presentation:** 1
**Contribution:** 2
**Rating:** 4
**Confidence:** 3

**Summary:**

The paper proposed a method to train a NeRF with blurry (due to camera motion), low-light scene images. After training the method allows the recovery of enhanced, sharp images. To solve the problem two modules are proposed:

1) A SND module for noise modeling, which includes both a noise-prediction NeRF (N-NeRF) and a consistency loss, which uses image matching to constrain the radiance to be multiview consistent

2) A CTP module to model the blur induced by camera motion

The paper evaluates the method and baselines on a self-collected dataset.

**Strengths:**

Originality: I appreciate the problem being tackled in the paper and think this is an underexplored problem. I think we are still far from being able to use NeRFs on casually captured phone videos, as evidenced by apps such as Luma, which although produce excellent quality require no (camera) motion blur and perfect lighting in the captures. I also appreciate the authors trying to assemble their own dataset although I do have some qualms about their claims (see weakness section).

Quality: I think the experiments and baselines have been chosen well.

**Weaknesses:**

Quality: Even though I believe the dataset to be useful, I find it hard to attribute the dataset to the authors of the paper. If I understand correctly (please correct me if I am wrong), the original images were actually captured by prior work (Lednet), and the authors of LuSh-NeRF only ran an off-the-shelf SFM method to recover the camera parameters? Happy to be shown otherwise, but I do not think in this case the authors can claim the dataset as a contribution of the paper.
Other than that although the results seem convincing, the contribution of the paper isn’t significant.

Clarity: I had serious issues understanding the paper, specifically the methods section. Authors introduce a lot of names, which I think would be better replaced by just mathematical notation. Some of the notation is also not properly explained, for example, $n_{N/2}$ in Eq. 5. (see also questions).

It is also unclear to me how exactly Equations 4 and 5 relate. If I understand correctly Eq 4. suggests that to render the low-light, blurry images the network prediction is first unsharpened and then noise is added, but in Eq 5. noise is directly added to the radiance predictions from the network?

I think part of the CTP module also comes from the paper DP-NeRF, but I think the paper does not exactly mention what this module does. In the interest of making the paper self-sufficient, I think the authors should elaborate on what the module does/what it is exactly. I think what they do elaborate on (the detaching of certain rays) is just their contribution on top of the method from DP-NeRF, if I understand correctly.


A few typos, there’s probably more I missed:

L145: bracket not closed

L183: bracket not opened

**Questions:**

1) I think the $n_{N/2}$ refers to the N-NeRF output at the 3D coordinate which is the midpoint of the ray samples, is that right? If we don’t care about the 3D consistency of N-NeRF (since noise is not 3D consistent) why is the noise even modeled with a NeRF? Why not just optimize a tensor per view, or have the noise be a ray-conditioned network (instead of 3D coordinate conditioned)?

2) I wonder how important it is to low-pass filter the images before thresholding for the CTP module, have the authors tried to just threshold the image without the low-pass filter?

**Limitations:**

Yes.

---

> ### Author Rebuttal · Authors · 2024-08-07
>
> ### W1: Insufficient Dataset Contribution.
> **R:** Actually, we did **not just** take images from the LOL-Blur dataset and then ran an off-the-shelf SFM method to obtain the camera parameters. To build an effective dataset, we did the following works:
>
>  (1) **Scene selection**: We went through the whole LOL-Blur dataset to select scenes **featuring different environments** (covering indoor/outdoor situations) with **different camera motions and lighting.**
>
>  (2) **Image selection**: We manually select 20-25 (out of around 60) images per scene with relatively **high-quality** for the estimation of camera poses, and ensure their **luminance** is close to imitate the real shooting situation.
>
>  (3) **Camera Pose Estimation**: Note that the estimation of camera pose by COLMAP for low-light blurry images are often **unreliable**. We first repeat the estimation of COLMAP **30 times for each scene** to select the optimal pose result, and then **manually tune** it with our baseline NeRF model to improve the accuracy.
>
> Our experimental results show that the resulting dataset helps us learn the **robust LuSh-NeRF** for handling low-light blurry scenes. We will clarify these in the revision.
>
> ### W2: Unclear notation.
> **R:** $n_{\frac{N}{2}}$ represents the noise value obtained by the N-NeRF MLP structure, and the $\frac{N}{2}$ means the middle sampled points on each ray.
>
> We thank this reviewer for pointing out this issue, and will **revise the Method section** to improve the clarity in the revision.
>
> ### W3: Relation of Equations 4 and 5.
> **R:** We apologize for the confusion caused by Eq.5, as it omits the deblur process in LuSh-NeRF, and the revised Eq.5 should be:
> $ C_{noisy}(r) = CTP(C_{S-NeRF}(r)) + C_{N-NeRF}(r) = CTP(\sum_{i=1}^{N} w_i c_i) + n_{\frac{N}{2}},
> \quad  where \quad n_{\frac{N}{2}} = MLP_{N-NeRF}(P_{mid}, d), $
>
> where the $P_{mid}$ and $d$ is the intermediate sampling points coordinate and the view direction. The $\text{CTP}(\cdot)$ function is the CTP module, which is illustrated in the Subsec. 3.3.
>
> ### W4: Relation between CTP module and DP-NeRF.
> **R:** The CTP module adopts the Rigid Blurring Kernel (RBK) module of DP-NeRF. The RBK module uses two MLPs to model the Rigid Camera Motion parameters for each viewpoint in the dense SE(3) field, one MLP to calculate the weights of the rendered pixels obtained from each camera pose within the kernel, then these pixels are used to compute the final blurry pixel values via weighted sum operation.
>
> The **difference** between the CTP and the RBK module lies in that, the **frequency-domain information** is introduced into the kernel estimation to help achieve **more accurate blur modeling** by exploiting the image regions which are less affected by the low light noise. We will clarity them in the revision.
>
> ### Q1: Why is the noise modeled with a NeRF? Why not optimize ...
> **R:** Thanks for the question. We refer to it N-NeRF as it is a multilayer MLP network **similar to the S-NeRF**. We will revise its name to **"Noise Estimator"** in the revision to avoid any ambiguities.
>
> The N-NeRF is **ray-conditioned** as the inputs are the view direction $d$ and the coordinates of the mid-point **(fixed for each ray)** during the whole training process.
>
> As suggested, we try the **view-dependent inputs** $(x,y,N)$, i.e., a 2D coordinate $(x,y)$ with the view id $N$ as the inputs to the N-NeRF to optimize a **noise tensor per view**. The results are reported below and the performance difference is **quite small**, which shows that **both strategies work** in our task.
>
> |Scene|"Dorm"|||"Poster"|||
> |-|-|-|-|-|-|-|
> ||PSNR$\uparrow$|SSIM$\uparrow$|LPIPS$\downarrow$|PSNR$\uparrow$|SSIM$\uparrow$|LPIPS$\downarrow$|
> |View-Independent Noise|**19.10**|0.5278|**0.3451**|18.08|**0.6348**|0.2343|
> |Ours|19.06|**0.5354**|0.3491|**18.12**|0.6331|**0.2265**|
>
> ### Q2 Part1: The importance of the low-pass filter.
> **R:** We have shown in the **Fig. 2 in rebuttal PDF** the difference between the two masks obtained by directly performing a rgb intensity threshold and performing a Frequency filtering before taking a threshold.
>
> (1) The RGB intensity of many **noise-dominant** regions are also **high** (e.g., the sky and the grass), which are harmful to the blur estimation but **not excluded in the mask** produced by thresholding.
>
> (2) The CTP module uses a Frequency filter to identify the **low-frequency-dominated** regions of the image, and then obtains the desired mask based on the RGB intensity values. The gradients of rays in high-frequency and/or dark regions (regions more **severely affected by noise**) are detached during the blur kernel optimization process, which ensures better blur modeling.
>
> ### Q2 Part2: Just threshold the image without the low-pass filter.
> **R:** We have conducted the ablation experiments of CTP module on two synthetic scenes in the following table.
> | Scene|"Dorm"||| "Poster" ||| Average |||
> |-|-|-|-|-|-|-|-|-|-|
> || PSNR$\uparrow$|SSIM$\uparrow$|LPIPS$\downarrow$|PSNR$\uparrow$|SSIM$\uparrow$|LPIPS$\downarrow$| PSNR$\uparrow$|SSIM$\uparrow$|LPIPS$\downarrow$|
> |No Threshold|18.99|0.5299|0.3630|18.05|0.6179|0.2598|18.52|0.5739|0.3114|
> |RGB Threshold(T=32)|**19.18**|0.5308|0.3580|17.86|0.5896|0.2450|18.52|0.5602|0.3015|
> |RGB Threshold(T=48)|18.71|0.4874|0.4455|17.92|0.6057|0.2418|18.32|0.5466|0.3437|
> |RGB Threshold(T=64)|18.62|0.4804|0.4474|17.53|0.5947|0.2420|18.08|0.5376|0.3447|
> |CTP Threshold(r=10, T=48)|18.95|0.5052|0.3660|18.14|0.6306|0.2275|18.55|0.5679|0.2968|
> |CTP Threshold(r=30, T=32)|19.02|0.5310|0.3515|**18.23**|0.6318|0.2373|**18.63**|0.5814|0.2944|
> |**CTP Threshold(r=30, T=48)**|19.06|**0.5354**|**0.3491**|18.12|**0.6331**|**0.2265**|18.59|**0.5843**|**0.2878**|
> |CTP Threshold(r=50, T=48)|19.10|0.5051|0.3634|17.78|0.6114|0.2384|18.44|0.5583|0.3009|
>
> The experimental results show that the frequency filter provides a **more desirable** mask, which is superior to the directly obtaining mask with the RBG intensity thresholds.

---

> ### Comment · Reviewer_4eod · 2024-08-11
>
> Thank you for the response to my comments.
>
>
> **W1: Insufficient Dataset Contribution.**
>
> For the discussion on the dataset, I am still not satisfied, I would not attribute the paper any significant contribution for selecting views for 5 synthetic and 5 captured scenes from an existing dataset and estimating the camera parameters, even though I appreciate COLMAP is not always straightforward to use.
>
>
> **W4: Relation between CTP module and DP-NeRF.**
>
> Thank you very much for this explanation, that helps me better understand the difference between the two modules. But frankly, I think even this explanation is not enough for what the RBK module exactly does, I think its common practice to explain properly the methods used if they are non-standard even if they come from other papers. I would implore the authors to add a more thorough description, even if just in the supplement.
>
>
> **Q1. Why is the noise modeled with a NeRF?**
>
> This begs the question, why the authors don't use the (x, y, N) based formulation. Isn't this formulation more efficient and understandable? Presumably, this MLP is smaller, and since there is no spatial consistency required, it makes more sense to use it than a 3D-conditioned MLP?
>
> EDIT: Maybe I am misunderstanding something, what is the ray midpoint the author's query? This changes as per the random sampling, is that right? So for the same pixel, I could be querying a different midpoint in different iterations?
>
> Otherwise, I thank the authors for their responses, I am satisfied with the answers provided to all other questions. I think the discussion on the low-pass filter is especially useful, since the Freq. Domain Thresholding is one of the main contributions to the CTP module. I would implore the authors to add these results to the paper.

---

> ### Author Response · Authors · 2024-08-12
>
> Thanks for your reply and we are glad to see that our response can address most of your raised concerns. We would like to futher clarify the below issues.
>
> ### R3W1: Insufficient Dataset Contribution.
> **R:** (1) The dataset contribution is one **part of** our **3rd contribution** and we still have **other technical contributions**.
>
> (2) Note that our dataset is necessary as we are handling a **new task** (handling NeRF in low-light scenes with camera motions). To construct this dataset, it took us more than 3 full weeks to select images and tune the camera pose parameters. Note that the colmap-free method [A] does not handle our task well as it is **very difficult** to optimize the camera pose directly in **low-light blurry** scenes.
>
> (3) We can **revise** our third contribution to **emphasize more** on the experimental evaluations and state-of-the-art results of our model. However, we **do need this dataset** for evaluation and we did put efforts into constructing this dataset.
>
> [A] NeRF--: Neural radiance fields without known camera parameters, arXiv:2102.07064, 2021.
>
>
>
> ### R3W4: Relation between CTP module and DP-NeRF.
> **R:** Thansk for your suggestion. The Rigid Blurring Kernel (RBK) module in DP-NeRF[19] models the scene blur kernel by simulating the 3D camera motions via the following two main parts:
>
> **Ray Rigid Transformation (RRT):** The RRT models the blurring process of an image. It is formulated as ray transformation derived from the deformation of rigid camera motion, which is defined as the dense SE(3) field for scene $s$ and modeled by the MLPs as:
>
> $S_s = (r_s;v_s) = (\mathcal{R}(\mathcal{E}(l_s);\mathcal{L}(\mathcal{E}(l_s))), where \  s \in view_{img},$
>
> where $l_s$ is the latent code for each view through the embedding layer in [B], $\mathcal{R}, \mathcal{L}, \mathcal{E}$ are three MLP networks, $view_{img}$ is the training view set. The $S_s = (r_s;v_s) \in \mathbb{R}^6$ is the matrix which will be used for the RRT modeling as follows:
>
> $ray_{s;q}^{RRT} = Rigid-Transform(ray_s, (r_s;v_s)),$
>
> where $Rigid-Transform()$ function is the standard 3D rigid transformation operation, $ray_s$ and $ray_{s;q}^{RRT}$ are the orgin ray and the transformed rays in scene $s$, $q \in \\{1,...,k \\}$, $k$ is a hyper-parameter that controls the number of camera motions contributing to the blur in each scene $s$. The blurry RGB value at $ray_s$ can be acquired by weighting sums of the NeRF volume rendering values $C_{s;0}$ and $C_{s;q}$ from $ray_s$ and $ray_{s;q}^{RRT}$.
>
> **Coarse Composition Weights:** The Coarse Composition Weights are computed for each ray obtained by the RRT:
>
> $m_{s;0,1,...,k} = \sigma(\mathcal{W}(\mathcal{E}(l_s))), where \  \sum_{i=0}^{k}m_{s;i} = 1,$
>
> where $m_s$ is the final weight for each ray in RRT. Finally, blurry color $\mathnormal{C}_s$ for scene $s$ can be computed by the weighted sum operation as shown below:
>
> $C_s = m_{s;0}C_{s;0} + \sum_{q=1}^{k}m_{s;q}C_{s;q}.$
>
> **We will add these information to the revision.**
>
> [B] Optimizing the latent space of generative networks, arXiv:1707.05776, 2017.
>
> ### R3Q1: Why is the noise modeled with a NeRF? Why not optimize ...
> **R:** We agree with the reviewer that the $(x,y,N)$-based formulation is another possbile implementation of modeling noise comparing to our current one (the rendering results of both **do not differ much**). Note that this **does not affect our goal** of decomposing the scene and noise information.
>
> Regarding the ray midpoint, we **uniformly sample (instead of random sampling in S-NeRF)** between the near and far fields (calculated by COLMAP) of the ray, and select the coordinates $P_{mid}$ of the intermediate sampling point, along with the view direction $d$, as the input of N-NeRF. Since the **bounds of the ray do not change** during the training phase, the midpoints for each ray (one-to-one map with pixels) is same.
>
> We will incorporate all these information into our revision.

---

> > ### Author Response · Authors · 2024-08-13
> >
> > Dear Reviewer 4eod,
> >
> > Thank you again for your review and reply. We hope that our rebuttal and the following comments could address your questions and concerns.  As the discussion phase is nearing its end, we would be grateful to hear your feedback and wondered if you might still have any concerns we could address.
> >
> > It would be appreciated if you could raise your score on our paper if we address your concerns. We thank you again for your effort in reviewing our paper.
> >
> > Best regards,
> >
> > Authors of Paper #10302

---

> > > ### Comment · Reviewer_4eod · 2024-08-14
> > >
> > > Thank you very much! I appreciate that the discussion is coming to an end, I don't think I need any more input from the authors. I'm not sure I will increase my score but I will relay my thoughts to the AC.
> > >
> > > **Method section.** Ultimately, I would just like to implore the authors to rewrite the methods section a bit more clearly. My suggestion would be to limit the use of these short forms such as RRT, CTP etc. I think it's difficult to keep track of all these.
> > >
> > > Maybe use just two short forms for the two main parts of the method (SND, CTP). And make separate paragraphs within sections to describe each component of these separate paths.
> > >
> > > **Dataset.** I would also de-emphasize the dataset claim a bit, as the authors suggested.
> > >
> > > Thanks a lot!

---

> > > > ### Author Response · Authors · 2024-08-14
> > > >
> > > > Thanks for your suggestions. We promise to **revise our Method section** in the revision to ensure better understanding.

---

### Official Review · Reviewer_54cf · 2024-07-12

**Soundness:** 3
**Presentation:** 2
**Contribution:** 3
**Rating:** 6
**Confidence:** 5

**Summary:**

The authors propose LuSh-NeRF, a model that reconstructs a clean and sharp NeRF from handheld low-light images by sequentially modeling noise and blur. LuSh-NeRF includes a Scene-Noise Decomposition (SND) module for noise removal and a Camera Trajectory Prediction (CTP) module for estimating camera motions based on low-frequency scene information. Experiments demonstrate that LuSh-NeRF outperforms existing methods in rendering bright and sharp novel-view images from low-light scenes.

**Strengths:**

1.The paper aims to address the challenge of using NeRFs in low-light conditions, where images often suffer from low visibility, noise, and camera shakes together.

2. A new dataset containing synthetic and real images is constructed to facilitate training and evaluation.

3. The method is grounded in a sound theoretical framework, leveraging multi-view consistency and frequency-domain information.

**Weaknesses:**

1. For me, the ideas and motivation behind this paper are quite good. What I am concern is whether the authors could consider adding some ablation studies, such as the role of various modules in LuSh-NeRF, this could let reader know which part is more effective.

2. In the scenario described by the author, I think the colmap-free NeRF method of this task may be more effective. Low-light blurred images have a certain impact on colmap estimation.

**Questions:**

1. I have another small suggestion. Could the authors consider compressing the images, or perhaps saving them in PDF format? This might help reduce the memory size of the paper's PDF file.

2. Could the author show some comparison with Aleth-NeRF [7] ? which is also a low-light NeRF method.

**Limitations:**

Please refer to the weakness part.

---

> ### Author Rebuttal · Authors · 2024-08-07
>
> ### W1: Ablation studies regarding the roles of proposed modules.
> **R:** Thanks for your positive feedback on our work, the **visualization** of the ablation experiments can be found in **Fig.6** in the main text. To better demonstrate the effectiveness of the different modules in LuSh-NeRF, we performed **quantitative ablation experiments** on all the synthetic datasets in the table below:
> | Scene | Dorm  |  |  | Poster |  |   | Plane |  |  | Sakura |  |   | Hall  |  | | Average |  |  |
> |------------------------------|-------|--------|---------|--------|---------|---------|-------|--------|--------|--------|---------|--------|-------|--------|---------|---------|---------|---------|
> |                              | PSNR$\uparrow$  | SSIM$\uparrow$   | LPIPS$\downarrow$   | PSNR$\uparrow$   | SSIM$\uparrow$    | LPIPS$\downarrow$   | PSNR$\uparrow$  | SSIM$\uparrow$   | LPIPS$\downarrow$  | PSNR$\uparrow$   | SSIM$\uparrow$    | LPIPS$\downarrow$  | PSNR$\uparrow$  | SSIM$\uparrow$   | LPIPS$\downarrow$   | PSNR$\uparrow$    | SSIM$\uparrow$    | LPIPS$\downarrow$   |
> | NeRF                         | 6.02  | 0.0307 | 0.8030  | 11.25  | 0.5159  | 0.4061  | 5.53  | 0.0716 | 0.8418 | 7.54   | 0.0553  | 0.7186 | 8.19  | 0.1679 | 0.5048  | 7.706   | 0.1683 | 0.6549 |
> | ScaleUp                 | 19.75 | 0.5599 | 0.4705  | 19.07  | 0.7048  | 0.3088  | 19.66 | 0.523  | 0.4986 | 18.73  | 0.5699  | 0.3666 | 21.34 | 0.7213 | 0.2831  | 19.71   | 0.6158 | 0.3855 |
> | ScaleUp+Blur Kernel          | 18.89 | 0.5259 | 0.4353  | 17.23  | 0.5900  | 0.2805  | 19.13 | 0.5193 | 0.4185 | 18.23  | 0.5482  | 0.2789 | 20.43 | 0.6353 | 0.2684  | 18.78  | 0.5637 | 0.3363 |
> | ScaleUp+CTP                  | 18.52 | 0.5205 | 0.3654  | 17.02  | 0.5915  | 0.2415  | 19.32 | 0.5144 | 0.4048 | 18.27  | 0.5514  | 0.2715 | 20.25 | 0.6411 | 0.2577  | 18.68  | 0.5638 | 0.3082 |
> | ScaleUp+SND                  | **20.18** | **0.5646** | 0.4490  | **21.37**  | **0.7542**  | 0.2534  | **20.13** | **0.5665** | 0.4873 | **19.16**  | **0.5889**  | 0.3568 | **21.67** | **0.7326** | 0.2801  | **20.50**  | **0.6414** | 0.3653 |
> | ScaleUp+Blur Kernel+SND      | 18.99 | 0.5299 | 0.3630  | 18.05  | 0.6179  | 0.2598  | 18.93 | 0.5191 | 0.3954 | 18.65  | 0.5530  | 0.2752 | 20.74 | 0.6381 | 0.2434  | 19.07  | 0.5716  | 0.3074 |
> | LuSh-NeRF (Sharp -> Denoise) | 18.66 | 0.5008 | 0.3514  | 17.38  | 0.5860   | 0.2600  | 19.13 | 0.5213 | 0.4076 | 18.24  | 0.5420  | 0.2589 | 20.72 | 0.6386 | 0.2667  | 18.83  | 0.5578 | 0.3089 |
> | LuSh-NeRF (Denoise -> Sharp) | 19.06 | 0.5354 | **0.3491**  | 18.12  | 0.6331  | **0.2265**  | 19.34 | 0.5275 | **0.3852** | 18.94  | 0.5884  | **0.2562** | 21.09 | 0.6421 | **0.2400**  | 19.31   | 0.5853  | **0.2914**  |
>
> The specific analysis is as follows:
>
> (1) From Line 1 and 2, the ScaleUp preprocessing enhances the NeRF's reconstruction capabilities, resulting in improved rendering results.
>
> (2) From Line 3 and 4, CTP leverages frequency domain information to refine Blur Kernel predictions, boosting the perceptual quality of reconstructed images. However, this process **diminishes** PSNR and SSIM results due to neglecting **noise interference**.
>
> (3) From Line 2 and 5, the SND module can **disentangle noise and scene** information from input noisy-blurry images, leading to substantial improvements in PSNR and SSIM metrics. However, the SND module is not capable of resolving the blur problem, which leads to an **minor improvement** in the image's perceptual quality (**more important** for the rendered images).
>
> (4) From Line 4, 5, 8, the combined application of SND and CTP modules enhances image **perceptual quality** and outperforms **the sole use** of CTP in terms of PSNR and SSIM and SND in terms of LPIPS.
>
> (5) From Line 7, 8, it can be concluded that **decoupling the noise first**, and **then modeling the scene blur** is a more robust restoration order, which can effectively reduce the interference of noise in the deblurring process, and obtain better performance metrics.
>
> ### W2: Colmap-free NeRF methods may be more effective?
> **R:** The authors highly agree with your ideas. Colmap-free NeRF methods robust to low-light and blur phenomena would be much more helpful for the problem we propose.  However, existing colmap-free NeRF methods may not handle our task easily.
>
> The table below compares to a COLMAP-Free NeRF [1]. The results demonstrate the existing COLMAP-free NeRF method cannot effectively handle low-light scenes with motion blur, due to the inaccurate poses optimized from the image directly. Exploring colmap-free methods for our task can be interesting for future work.
>
> | Scene| "Dorm" | || "Poster" | | | "Plane" | | | Average | | |
> |--|--|--|--|--|--|--|--|--|--|--|--|--|
> || PSNR$\uparrow$ | SSIM$\uparrow$| LPIPS$\downarrow$ | PSNR$\uparrow$ | SSIM$\uparrow$| LPIPS$\downarrow$ | PSNR$\uparrow$  | SSIM$\uparrow$ | LPIPS$\downarrow$| PSNR$\uparrow$ | SSIM$\uparrow$| LPIPS$\downarrow$|
> | Preprocess+NeRF(COLMAP) | **19.75** | **0.5599**| **0.4705** | 19.07| **0.7048** | **0.3088** | **19.66**   | 0.5230  | **0.4986** | **19.49** | **0.5959** | **0.4260** |
> | Preprocess+NeRF-\-[1]  | 18.95  | 0.5423 | 0.4762 | **19.13** | 0.6935 | 0.3341 |19.62| **0.5243** | 0.5129 | 19.23 | 0.5867 | 0.4411 |
>
> [1] Wang Z, Wu S, Xie W, et al. NeRF--: Neural radiance fields without known camera parameters[J]. arXiv preprint arXiv:2102.07064, 2021.
>
> ### Q1: Reduce the memory size of the paper.
> **R:** Thanks for the suggestion, we will compress the images and reduce the PDF size in our revision.
>
> ### Q2: More comparison with Aleth-NeRF.
> **R:** As suggested, we have performed several experiments with Aleth-NeRF [7] on our proposed synthetic and realistic scenes. The visualization results of one comparison in this experiment are shown **in the Fig.1 in Rebuttal PDF**.
>
> As Aleth-NeRF focuses on adjusting the luminance of the scene, it **cannot handle the blur** in low-light scenes. Will include more in the revision.

---

> ### Comment · Reviewer_54cf · 2024-08-08
>
> Thank you for your response. My concerns have been fully resolved, and this is an excellent work. I improve my rank to weak accept.

---

> > ### Author Response · Authors · 2024-08-10
> >
> > We're very glad that our rebuttal has addressed your concerns, and thank you for recognizing the paper, we'll add the missing details in the revision.

---

### Official Review · Reviewer_4gW7 · 2024-07-13

**Soundness:** 3
**Presentation:** 3
**Contribution:** 3
**Rating:** 6
**Confidence:** 4

**Summary:**

This method proposes a solution for NeRF optimization under low light settings by resolving 3 different forms of degradation: low intensity, camera noise, and motion blur. Low intensity is effectively resolved by scaling up the image, camera noise is resolved by proposing a consistency loss between different views to locate the noise, and motion blur is handled by using a camera trajectory prediction module to predict the camera motion and thus the corresponding sharpening function. The experiments demonstrate state-of-the-art performance on novel view synthesis given low-light conditions. Qualitative ablations are provided to support the contribution of each component of the method. The authors will also make their code readily available as well as a dataset with 5 synthetic and 5 real scenes for evaluating low-light novel view synthesis.

**Strengths:**

The experiments support that the model is state-of-the-art, as in most scenes the proposed method achieves the best performance for low-light novel view synthesis.

The model is well thought out and handles many types of degradations, including low intensity, noise, and motion blur. The SND and CTP modules are well designed and novel components of the method.

The paper presentation is clear and easy to follow with appropriate figures to aid the reader's understanding.

The code will be made publicly available and a dataset will be released to aid evaluations in the low-light novel view synthesis domain.

**Weaknesses:**

The ablations would be more convincing with some quantitative results to back up the qualitative results.

There appear to be some mistakes in Table 1: in some columns two numbers are underlined, in others the best and second best seem to be incorrectly reported. Please be careful to fix these mistakes.

There are some important citations missing for some parts of the related work, such as in line 71 when mentioning NeRF methods that model digital humans:

1. HeadNeRF (CVPR 2022)
2. MoFaNeRF (ECCV 2022)
3. INFAMOUS-NeRF (arxiv, 2023)

**Questions:**

As mentioned above, please provide quantitative results to support the ablation studies and be sure to fix the errors in Table 1. I will be considering the rebuttal carefully as well as the thoughts from other reviewers in deciding my final rating.

**Limitations:**

Limitations and broader impact section are both available and appropriate. This work naturally does not have many immediately obvious negative societal impacts as it is simply trying to faithfully render novel-view images in low-light conditions.

---

> ### Author Rebuttal · Authors · 2024-08-07
>
> ### W1: Ablation studies with quantitative results.
> **R:** As suggested, we have conducted detailed ablation studies on all the synthetic scenes in the following table:
> | Scene                        | Dorm  |        |         | Poster |         |         | Plane |        |        | Sakura |         |        | Hall  |        |         | Average |         |         |
> |------------------------------|-------|--------|---------|--------|---------|---------|-------|--------|--------|--------|---------|--------|-------|--------|---------|---------|---------|---------|
> |                              | PSNR$\uparrow$  | SSIM$\uparrow$   | LPIPS$\downarrow$   | PSNR$\uparrow$   | SSIM$\uparrow$    | LPIPS$\downarrow$   | PSNR$\uparrow$  | SSIM$\uparrow$   | LPIPS$\downarrow$  | PSNR$\uparrow$   | SSIM$\uparrow$    | LPIPS$\downarrow$  | PSNR$\uparrow$  | SSIM$\uparrow$   | LPIPS$\downarrow$   | PSNR$\uparrow$    | SSIM$\uparrow$    | LPIPS$\downarrow$   |
> | NeRF                         | 6.02  | 0.0307 | 0.8030  | 11.25  | 0.5159  | 0.4061  | 5.53  | 0.0716 | 0.8418 | 7.54   | 0.0553  | 0.7186 | 8.19  | 0.1679 | 0.5048  | 7.706   | 0.1683 | 0.6549 |
> | ScaleUp                 | 19.75 | 0.5599 | 0.4705  | 19.07  | 0.7048  | 0.3088  | 19.66 | 0.5230  | 0.4986 | 18.73  | 0.5699  | 0.3666 | 21.34 | 0.7213 | 0.2831  | 19.71   | 0.6158 | 0.3855 |
> | ScaleUp+Blur Kernel          | 18.89 | 0.5259 | 0.4353  | 17.23  | 0.5900  | 0.2805  | 19.13 | 0.5193 | 0.4185 | 18.23  | 0.5482  | 0.2789 | 20.43 | 0.6353 | 0.2684  | 18.78  | 0.5637 | 0.3363 |
> | ScaleUp+CTP                  | 18.52 | 0.5205 | 0.3654  | 17.02  | 0.5915  | 0.2415  | 19.32 | 0.5144 | 0.4048 | 18.27  | 0.5514  | 0.2715 | 20.25 | 0.6411 | 0.2577  | 18.68  | 0.5638 | 0.3082 |
> | ScaleUp+SND                  | **20.18** | **0.5646** | 0.4490  | **21.37**  | **0.7542**  | 0.2534  | **20.13** | **0.5665** | 0.4873 | **19.16**  | **0.5889**  | 0.3568 | **21.67** | **0.7326** | 0.2801  | **20.50**  | **0.6414** | 0.3653 |
> | ScaleUp+Blur Kernel+SND      | 18.99 | 0.5299 | 0.3630  | 18.05  | 0.6179  | 0.2598  | 18.93 | 0.5191 | 0.3954 | 18.65  | 0.5530  | 0.2752 | 20.74 | 0.6381 | 0.2434  | 19.07  | 0.5716  | 0.3074 |
> | LuSh-NeRF (Sharp -> Denoise) | 18.66 | 0.5008 | 0.3514  | 17.38  | 0.5860   | 0.2600  | 19.13 | 0.5213 | 0.4076 | 18.24  | 0.5420  | 0.2589 | 20.72 | 0.6386 | 0.2667  | 18.83  | 0.5578 | 0.3089 |
> | LuSh-NeRF (Denoise -> Sharp) | 19.06 | 0.5354 | **0.3491**  | 18.12  | 0.6331  | **0.2265**  | 19.34 | 0.5275 | **0.3852** | 18.94  | 0.5884  | **0.2562** | 21.09 | 0.6421 | **0.2400**  | 19.31   | 0.5853  | **0.2914**  |
>
> (1) From Line 1 and 2, the ScaleUp preprocessing enhances the NeRF's reconstruction capabilities, resulting in improved image.
>
> (2) From Line 3 and 4, CTP leverages frequency domain information to refine Blur Kernel predictions, boosting the perceptual quality of reconstructed images. However, this process **diminishes** PSNR and SSIM results due to neglecting **noise interference**.
>
> (3) From Line 2 and 5, the SND module can **disentangle noise and scene** information from input noisy-blurry images, leading to substantial improvements in PSNR and SSIM metrics. However, the SND module is not capable of resolving the blur problem, which leads to an **minor improvement** in the image's perceptual quality (**more important** for the rendered images).
>
> (4) From Line 4, 5, 8, the combined application of SND and CTP modules enhances image **perceptual quality** and outperforms the sole use of CTP in terms of PSNR and SSIM and SND in terms of LPIPS.
>
> (5) From Line 7, 8, it can be concluded that **decoupling the noise first**, and **then modeling the scene blur** is a more robust restoration order, which can effectively reduce the interference of noise in the deblurring process, and obtain better performance metrics.
>
> ### W2: Mistakes in Tab.1.
> **R:** Thanks for your careful reading, we will correct these mistakes **in the revision**.
>
>
> ### W3: Missing Citations.
> **R:** Thanks for your valuable reviews. HeadNeRF [1] integrates NeRF to the parametric representation of the human head. MoFaNeRF [2] propose the first parametric model that maps free-view facial images into a vector space with NeRF. INFAMOUS-NeRF [3] propose a novel photometric surface constraint that improves the face rendering performance.
>
> These inspiring works are of importance to the Human NeRF development. As suggested, we will **cite and discuss** these works in our revision.
>
> [1] Hong, Yang, et al. "Headnerf: A real-time nerf-based parametric head model." Proceedings of the IEEE/CVF Conference on Computer Vision and Pattern Recognition. 2022.
>
> [2] Zhuang, Yiyu, et al. "Mofanerf: Morphable facial neural radiance field." European conference on computer vision. Cham: Springer Nature Switzerland, 2022.
>
> [3] Hou, Andrew, et al. "INFAMOUS-NeRF: ImproviNg FAce MOdeling Using Semantically-Aligned Hypernetworks with Neural Radiance Fields." arXiv preprint arXiv:2312.16197 (2023).

---

> > ### Comment · Reviewer_4gW7 · 2024-08-09
> >
> > Thanks for your thorough rebuttal! Please include these missing details in the final version (especially the ablation results) to help deliver a more convincing set of experiments. I will maintain my Weak Accept rating as I see most other reviewers are in agreement.

---

> > > ### Author Response · Authors · 2024-08-10
> > >
> > > Thanks a lot for your acknowledgement and valuable comments, we promise to incorporate all these missing details in the revision.

---

### Author Rebuttal · Authors · 2024-08-07

We thank all reviewers for their comments and suggestions. We are glad to see that reviewers comment our idea/work as novel (4gW7), sound (54cf), appreciated (4eod), and appealing (m3Lt). We address the raised concerns below and will revise our paper according to all comments. Please let us know if further clarification / discussion is needed.

---

### Decision · Program_Chairs · 2024-09-25

**Decision:**

Accept (poster)

**Comment:**

Three reviewers recommended a weak accept for this paper, while only one reviewer rated it as borderline reject. The primary concern of the latter reviewer was the clarity of the paper, though they acknowledged its originality, aligning with the views of the other reviewers. I encourage the authors to incorporate the reviewers' feedback in the final version.